# Impaired iron recycling from erythrocytes is an early hallmark of aging

Patryk Slusarczyk[1][†], Pratik Kumar Mandal[1][†], Gabriela Zurawska[1], Marta Niklewicz[1], Komal Chouhan[1], Raghunandan Mahadeva[1], Aneta Jończy[1], Matylda Macias[1], Aleksandra Szybinska[1], Magdalena Cybulska-Lubak[2], Olga Krawczyk[2], Sylwia Herman[3], Michal Mikula[2], Remigiusz Serwa[4,5], Małgorzata Lenartowicz[3], Wojciech Pokrzywa[1], Katarzyna Mleczko-Sanecka[1]*

[1]International Institute of Molecular and Cell Biology in Warsaw, Warsaw, Poland; [2]Maria Sklodowska-Curie National Research Institute of Oncology, Warsaw, Poland; [3]Laboratory of Genetics and Evolution, Institute of Zoology and Biomedical Research, Jagiellonian University, Cracow, Poland; [4]IMol Polish Academy of Sciences, Warsaw, Poland; [5]ReMedy International Research Agenda Unit, IMol Polish Academy of Sciences, Warsaw, Poland

*For correspondence:
kmsanecka@iimcb.gov.pl

[†]These authors contributed equally to this work

**Abstract** Aging affects iron homeostasis, as evidenced by tissue iron loading and anemia in the elderly. Iron needs in mammals are met primarily by iron recycling from senescent red blood cells (RBCs), a task chiefly accomplished by splenic red pulp macrophages (RPMs) via erythrophagocytosis. Given that RPMs continuously process iron, their cellular functions might be susceptible to age-dependent decline, a possibility that has been unexplored to date. Here, we found that 10- to 11-month-old female mice exhibit iron loading in RPMs, largely attributable to a drop in iron exporter ferroportin, which diminishes their erythrophagocytosis capacity and lysosomal activity. Furthermore, we identified a loss of RPMs during aging, underlain by the combination of proteotoxic stress and iron-dependent cell death resembling ferroptosis. These impairments lead to the retention of senescent hemolytic RBCs in the spleen, and the formation of undegradable iron- and heme-rich extracellular protein aggregates, likely derived from ferroptotic RPMs. We further found that feeding mice an iron-reduced diet alleviates iron accumulation in RPMs, enhances their ability to clear erythrocytes, and reduces damage. Consequently, this diet ameliorates hemolysis of splenic RBCs and reduces the burden of protein aggregates, mildly increasing serum iron availability in aging mice. Taken together, we identified RPM collapse as an early hallmark of aging and demonstrated that dietary iron reduction improves iron turnover efficacy.

## Editor's evaluation

Slusarczyk et al. present a well written manuscript focused on understanding the mechanisms underlying aging of erythrophagocytic macrophages in the spleen (RPM) and its relationship to iron loading with age. Importantly, the manuscript demonstrates that RPM erythrophagocytic capacity is diminished with age, restored in iron restricted diet fed aged mice. The main conclusion of the manuscript points to accumulation of unavailable insoluble forms of iron as both causing and resulting from RPM failure, likely a consequence of decreased ferroportin expression on RPMs in the spleen.

## Introduction

Sufficient iron supplies are critical for the proper functioning of cells and organisms (*Cronin et al., 2019*; *Muckenthaler et al., 2017*). At the systemic level, 80% of circulating iron is utilized for hemoglobin synthesis during the daily generation of approximately 200 billion red blood cells (RBCs) (*Muckenthaler et al., 2017*). The oxygen-carrying capacity of RBCs renders them sensitive to the progressive build-up of reactive oxygen species (ROS) that drive natural RBCs senescence (*Bratosin et al., 1998*). Due to the prooxidative properties of free heme and iron, high hemoglobin content in physiologically senescent RBCs constitutes a threat to tissues. To reduce the risk of RBC breakdown in the blood vessels and because mammals evolved under limited dietary iron availability, 90% of the body iron needs are met by internal iron recycling from aged RBCs (*Ganz, 2012*). This task is accomplished predominantly by red pulp macrophages (RPMs) of the spleen, cells that engulf defective RBCs in the process called erythrophagocytosis (EP; *Bian et al., 2016*; *Youssef et al., 2018*). Recent findings imply that in parallel to phagocytosis of intact senescent RBCs, which is actively performed by a fraction of RPMs at a given moment (*Akilesh et al., 2019*; *Ma et al., 2021*; *Youssef et al., 2018*) part of the RBC-derived iron is recovered via hemolysis of RBCs within the splenic microenvironment and subsequent hemoglobin uptake (*Klei et al., 2020*). The latter likely occurs in RPMs, but whether other cell types contribute to this process remains to be determined.

The loss of elasticity due to a build-up of oxidative damage is a key feature of naturally aged RBCs (*Arashiki et al., 2013*; *Bratosin et al., 1998*; *Ganz, 2012*; *Higgins, 2015*; *Lutz, 2012*). The unique architecture of the spleen confers a filter for verifying the biomechanical integrity of RBCs, whereby senescent rigid RBCs are retained within the spleen cords (*Mebius and Kraal, 2005*). Recognition of 'trapped' RBCs by RPMs involves additional mechanisms, such as the binding of exoplasmic phosphatidylserine or opsonizing antibodies, that were proposed to act additively (*Gottlieb et al., 2012*; *Slusarczyk and Mleczko-Sanecka, 2021*). Upon engulfment by RPMs, RBCs are degraded in phagolysosomes, and heme is released to the cytoplasm by HRG1 (*Klei et al., 2017*; *Pek et al., 2019*). Heme is subsequently catabolized by heme oxygenase 1 (HO-1; encoded by *Hmox1*) to carbon monoxide (CO), biliverdin, and ferrous iron that feeds the labile iron pool (LIP) (*Kovtunovych et al., 2010*). Iron from the LIP is sequestered by the iron-storing protein ferritin, composed of H and L subunits (*Mleczko-Sanecka and Silvestri, 2021*). Iron efflux from RPMs occurs via ferroportin (FPN) to replenish the transferrin-bound iron pool in the plasma (*Muckenthaler et al., 2017*; *Zhang et al., 2011*). The process of iron release from RPMs is tightly regulated by hepcidin, a liver-derived hormone that mediates FPN degradation and/or occlusion, hence preventing iron release from RPMs (*Aschemeyer et al., 2018*; *Nemeth et al., 2004*). Despite the growing body of knowledge, it is still not completely explored how iron balance in RPMs is affected by different pathophysiological conditions and how it may influence their functions.

Similar to erythropoiesis, the rate of RBC clearance is very high, reaching 2–3 million per second in humans (*Higgins, 2015*), and is considered largely constitutive under physiological conditions. Some reports showed that the intensity of EP can be enhanced by proinflammatory conditions and pathogen components (*Bennett et al., 2019*; *Bian et al., 2016*; *Delaby et al., 2012*). Recent work employing genetic mouse models showed that altered calcium signaling in RPMs due to overactivation or the loss of the PIEZO1 mechanoreceptor controls EP efficiency (*Ma et al., 2021*). However, it is largely unknown if the capacity for RBC uptake and their degradation within the phagolysosomes are regulated by intrinsic or systemic iron status per se.

Iron dyshomeostasis hallmarks physiological aging. This is exemplified by progressive iron accumulation and iron-dependent oxidative damage in aging organs (*Arruda et al., 2013*; *Cook and Yu, 1998*; *Sukumaran et al., 2017*; *Xu et al., 2008*). At the same time, limited plasma iron availability is frequent in aged individuals and is a leading cause of a condition referred to as anemia in the elderly (*Girelli et al., 2018*). RPMs are derived from embryonic progenitor cells (*Yona et al., 2013*), exhibit low self-renewal capacity (*Hashimoto et al., 2013*), and are only partially replenished by blood monocytes during aging (*Liu et al., 2019*). Hence, these specialized cells continuously perform senescent RBCs clearance during their lifespan, thus processing and supplying the majority of the organism's iron requirements. We hypothesized that the exposure to high iron burden might accelerate their aging, affecting body iron indices and RBC homeostasis. Therefore, we aimed to explore the exact relationship between substantial splenic iron deposition during early aging (*Altamura et al., 2014*) and the iron-recycling functions of RPMs. Here, we found that aging expands iron accumulation in RPMs

and their partial demise in female mice, largely attributable to the downregulation of ferroportin. We propose that the depletion of RPMs depends both on iron-triggered toxicity, resembling ferroptosis, and on the cell death mechanisms involving proteostasis defects. Furthermore, we demonstrate that iron loading impairs the capacity of RPMs to engulf and lyse RBCs. We show that age-related RPM dysfunction affects local RBC balance in the spleen, and RPM decay leads to the formation of iron- and heme-rich extracellular protein aggregates. Finally, we provide evidence that reducing dietary iron during aging 'rejuvenates' RPM functions and fitness, and improves iron homeostasis.

## Results

### RPMs of aged mice show increased labile iron levels, oxidative stress and diminished iron-recycling functions

To investigate iron-recycling capacity during aging, we used young (2–3 month-old) and aged (10–11 month-old) C57BL/6 J mice fed a diet with standard iron content (200 ppm). The latter age in mice corresponds to approximately 40–45 years in humans when senescent changes begin to occur (*Fox et al., 2006*). We used female mice which show more pronounced iron deposition in the spleen than males with age progression (*Altamura et al., 2014*). As expected, aged mice showed tissue iron loading, with the spleen being most affected, followed by the liver and other organs such as muscles and the heart (*Figure 1—figure supplement 1A–D*). Aging mice exhibited decreased blood hemoglobin values, as previously shown (*Peters et al., 2008*), and a drop in transferrin saturation, a feature that thus far was observed mostly in elderly humans (*Figure 1—figure supplement 1E, F*; *Girelli et al., 2018*). Next, we aimed to verify if aging affects the iron status and essential cellular functions of RPMs. Using intracellular staining and flow cytometric analyses, we uncovered that RPMs [gated as CD11b-dim, F4/80-high, TREML4-high (*Haldar et al., 2014*) as shown in *Figure 1—figure supplement 2*] in aged mice exhibited a significant deficiency in H ferritin levels, with unchanged L ferritin protein expression (*Figure 1—figure supplement 1G, H*; see *Figure 1—figure supplement 3A, B* for validation of antibodies for intracellular staining). Consistently, using the fluorescent probe Ferro-Orange that interacts specifically with ferrous iron, we detected a significant increase in LIP in aged RPMs (*Figure 1—figure supplement 1I*) accompanied by marked oxidative stress (*Figure 1—figure supplement 1J*). We next tested if this increase in labile iron in RPMs would impact the phagocytic activity. Hence, we incubated splenic single-cell suspension with temperature-stressed RBCs fluorescently labeled with PKH67 dye (*Klei et al., 2020*; *Theurl et al., 2016*). In parallel, we employed another standard cargo for phagocytosis, zymosan, a yeast cell wall component. Using this *ex vivo* approach, we detected a significant drop in RBC clearance capacity in aged compared to young RPMs (*Figure 1—figure supplement 1K*). Notably, their capacity for the engulfment of zymosan remained unchanged (*Figure 1—figure supplement 1L*), thus suggesting a specific defect of EP in aged RPMs. RPMs rely on lysosomal activity to lyse RBC components in phagolysosomes and hence exert their iron-recycling functions. Using a fluorescent probe, we observed decreased lysosomal activity in RPMs isolated from 10- to 11-month-old mice compared to those derived from young control animals (*Figure 1—figure supplement 1M*). Of note, we also detected diminished mitochondrial activity, a well-established aging marker (*Sun et al., 2016*), in aged RPMs (*Figure 1—figure supplement 1N*). Interestingly, peritoneal macrophages of aged mice did not show altered ROS levels or impaired functions of lysosomes and mitochondria (*Figure 1—figure supplement 4*), suggesting that these age-related changes affect RPMs earlier than other macrophage populations. Altogether, these insights suggest that during aging, RPMs increase LIP and exhibit a reduced capacity for the RBCs engulfment and lysosomal degradation of erythrocyte components.

### Iron-reduced diet normalizes body iron parameters during aging, diminishes iron retention in RPMs, and rescues age-related decline in RPMs' iron-recycling capacity

We next set out to explore whether an age-related alteration of systemic iron indices could be prevented. Caloric restriction was previously shown to effectively reduce tissue iron content in older rats (*Cook and Yu, 1998*; *Xu et al., 2008*). Analogously, to correct iron dyshomeostasis in aging mice, we reduced dietary iron content from the fifth week of life to a level (25 ppm) that was reported as sufficient to maintain erythropoiesis (*Sorbie and Valberg, 1974*). This approach reverted the degree

of iron deposition in the spleen and liver (*Figure 1A and B*) and partially alleviated the decreased transferrin saturation characteristic of aged mice fed a standard diet (*Figure 1C*). Notably, the mild drop in blood hemoglobin was not rescued by the iron-reduced (IR) diet (*Figure 1—figure supplement 5A*). Of note, aged mice did not display significant changes in hematocrit, RBCs counts, MCV, MCH, or the intensity of bone marrow erythropoiesis (*Figure 1—figure supplement 5B–F*; see *Figure 1—figure supplement 6* for gating strategy of erythroid progenitors). We observed a small increase in extramedullary erythropoietic activity in the spleen and increased plasma erythropoietin (EPO) levels in aged mice, implying that very mild anemic phenotype and/or decreased transferrin saturation are sensed by the body (*Figure 1—figure supplement 5G, H*). However, these responses were not diminished by the reduced dietary iron content in aged mice. These observations imply that either partial alleviation of plasma iron availability in IR mice is insufficient to correct very mild aging-triggered anemia or other hematopoiesis-related mechanisms may contribute to this phenotype.

Next, we observed that hepcidin mRNA levels were increased in the liver of aged *versus* young mice, a phenotype that was reverted by feeding them an IR diet (*Figure 1D*). In addition, we did not observe a contribution of inflammatory cues to hepcidin regulation, as the plasma IL-6 levels remained unchanged (*Figure 1E*). We next examined FPN levels in RPMs using an antibody that recognizes the extracellular loop of murine FPN (validated in *Figure 1—figure supplement 7*). As expected from the pattern of hepcidin expression, cell membrane FPN levels in RPMs were decreased in aged mice and this was rescued by feeding them an IR diet (*Figure 1F*). We further characterized RPMs' iron status. We found that despite decreased ferritin H levels in aged mice, regardless of the dietary regimen (*Figure 1G*), accumulation of both labile iron (*Figure 1H*) and total iron (*Figure 1I*) was completely reversed in RPMs of aged mice fed an IR diet as compared to a standard diet. Likewise, we observed alleviation of oxidative stress in aged RPMs in response to a dietary iron restriction (*Figure 1J*).

We next assessed whether altered RPM iron content in aged mice that were maintained on a standard affects their functions. First, we detected a lower representation of RPMs in the spleens of aged mice, a phenotype that was alleviated by limited dietary iron content (*Figure 1K*). We noticed that among 55 genes that were differentially regulated in RNA sequencing (RNA-seq) data between standard and IR diets, 9 genes are involved in proliferation control (*Figure 1—figure supplement 8A*). Consistent with the pattern of their regulation, we found that RPMs derived from aged mice on an IR diet showed mildly increased levels of the proliferation marker Ki-67 (*Figure 1—figure supplement 8B*), a phenomenon that was previously reported to drive RPM niche replenishment after erythrophagocytosis-driven depletion (*Youssef et al., 2018*). Next, by reanalysis of published data from Ms4a3-tdTomato reporter mice (*Liu et al., 2019*) (where macrophages of monocyte origin are fluorescently labeled) we demonstrated that the shortage of RPMs during aging is chiefly driven by the depletion of RPMs of embryonic origin, whereas the low rate of RPM replenishment from monocytes remained constant during aging (*Figure 1—figure supplement 9A*). In line with this, we did not observe any robust change in the splenic population of monocytes or pre-RPMs (*Lu et al., 2020*) in our aged mice (*Figure 1—figure supplement 9B, C*), and we did not detect significant differences in the numbers of granulocytes (*Figure 1—figure supplement 9D*). These data suggest that the drop in RPM representation is not underlain by insufficient recruitment from monocytes or a concurrent increase in granulocyte representation, but involves the demise of embryonically derived cells.

We further assessed if RPM iron status during aging reflects their capacity to engulf and degrade RBCs. To this end, we first performed *in vivo* EP assay via transfusion of PKH67-stained temperature-stressed erythrocytes (*Lu et al., 2020*; *Theurl et al., 2016*). We observed that aged RPMs were less efficient in sequestering transfused RBCs, an impairment that was partially rescued by feeding mice an IR diet (*Figure 1L*). Similarly to the EP rate, we found that RPMs isolated from aged IR mice restored the lysosomal activity to the levels observed in young mice (*Figure 1M*). These data imply that the diminished capacity of RPMs to engulf and lyse RBCs in phagolysosomes during aging can be ameliorated by limiting dietary iron content. Since earlier studies demonstrated that under conditions of RPM impairment, iron-recycling functions are significantly supported by liver macrophages (*Theurl et al., 2016*), we also measured their EP capacity during aging. Using the transfusion of PKH67-stained RBCs we did not detect significant differences in the percentage of RBC-positive hepatic macrophages in aged mice (*Figure 1N*). However, consistently with the *de novo* recruitment of iron-recycling myeloid cells to the liver upon erythrocytic stress (*Theurl et al., 2016*), we noticed a significant expansion of the KC-like population in aged mice but not in the aged IR group (*Figure 1O*). This

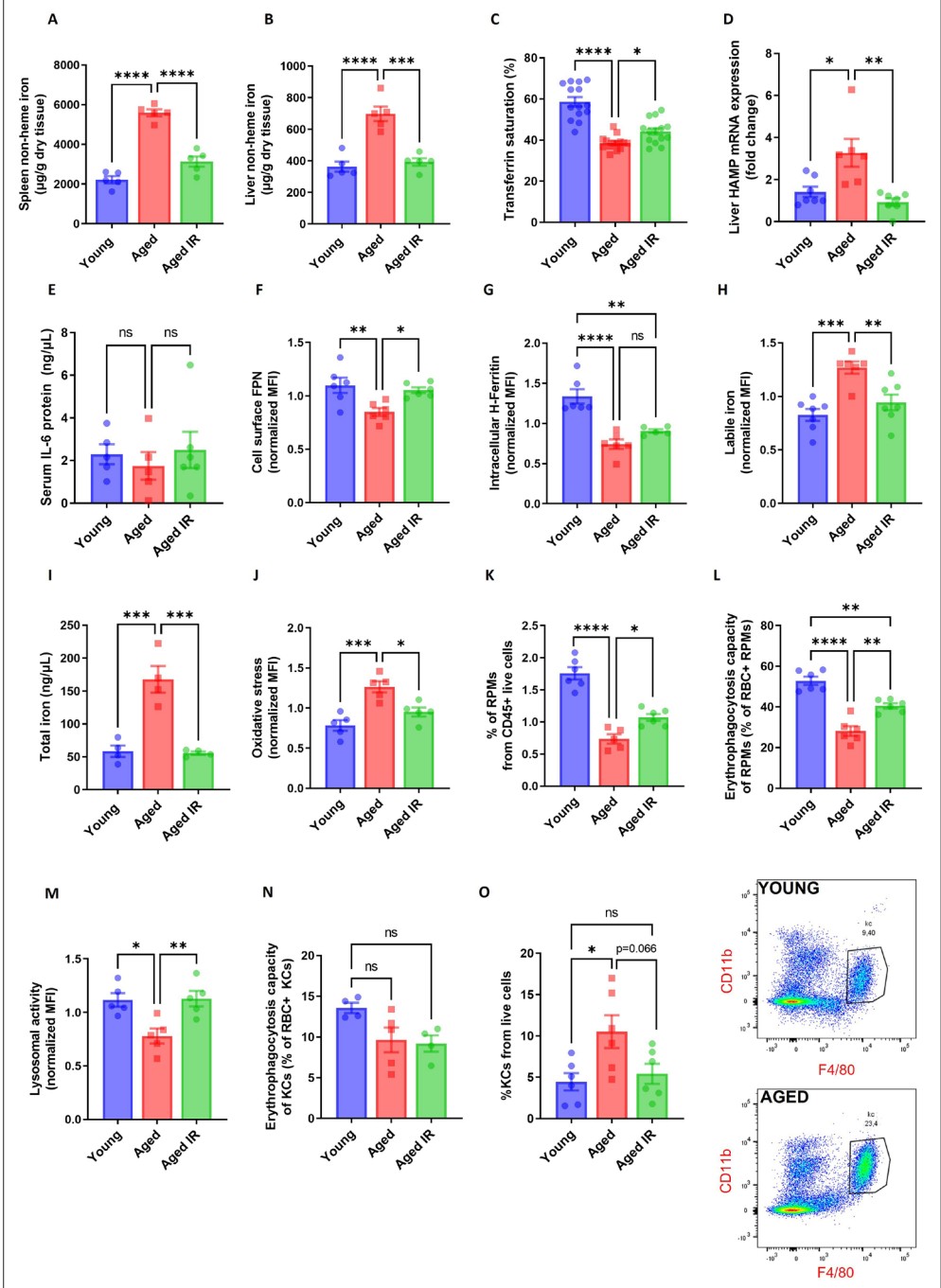

**Figure 1.** Iron-reduced diet normalizes body iron parameters during aging, diminishes iron retention in RPMs, and prevents oxidative stress. (**A**) Splenic and (**B**) liver non-heme iron content was determined in young, aged, and aged IR mice. (**C**) Plasma transferrin saturation was determined in young, aged, and aged IR mice. (**D**) Relative mRNA expression of hepcidin (Hamp) in the liver of young, aged, and aged IR mice was determined by qPCR. (**E**) Serum IL-6 protein levels in young, aged, and aged IR mice were measured by Mouse IL-6 Quantikine ELISA Kit. (**F**) Expression of ferroportin (FPN) on the cell membrane of young, aged, and aged IR RPMs was assessed by flow cytometry. (**G**) Intracellular H-Ferritin protein levels in young, aged, and aged IR RPMs were quantified by flow cytometry. (**H**) Cytosolic ferrous iron (Fe2+) levels in young, aged, and aged IR RPMs were measured using FerroOrange with flow cytometry. (**I**) The total intracellular iron content in young, aged, and aged IR magnetically-sorted RPMs was assessed using the Iron Assay Kit. (**J**) The cytosolic ROS levels in young, aged, and aged IR RPMs were assessed by determining CellROX Deep Red fluorescence intensity with flow cytometry. (**K**) The percentage of RPMs from CD45 + live cells present in the spleen of young, aged, and aged IR mice was

*Figure 1 continued on next page*

*Figure 1 continued*

assessed by flow cytometry. (**L**) Erythrophagocytosis capacity in young, aged, and aged IR RPMs was determined using flow cytometry by measuring the percentage of RPMs that phagocytosed transfused PKH67-labeled temperature-stressed RBCs. (**M**) Lysosomal activity of young, aged, and aged IR RPMs was determined using Lysosomal Intracellular Activity Assay Kit with flow cytometry. (**N**) Erythrophagocytosis capacity in young, aged, and aged IR Kupffer cells (KCs) was determined using flow cytometry by measuring the percentage of KCs that phagocytosed transfused PKH67-labeled temperature-stressed RBCs. (**O**) Percentages of KCs in total live cells in the livers of young, aged, and aged IR mice and representative flow cytometry plots of KCs. Each dot represents one mouse. Data are represented as mean ± SEM. Statistical significance among the three groups was determined by One-Way ANOVA test with Tukey's Multiple Comparison test. ns $p > 0.05$, *$p < 0.05$, **$p < 0.01$, ***$p < 0.001$ and ****$p < 0.0001$.

The online version of this article includes the following source data and figure supplement(s) for figure 1:

**Source data 1.** Related to *Figure 1A–O*.

**Figure supplement 1.** RPMs of aged mice show increased labile iron levels, oxidative stress and diminished iron-recycling functions.

**Figure supplement 1—source data 1.** Related to *Figure 1—figure supplement 1A–N*.

**Figure supplement 2.** Gating strategy for RPMs.

**Figure supplement 3.** Validation of the antibodies against H and L ferritin.

**Figure supplement 3—source data 1.** Related to *Figure 1—figure supplement 3A*.

**Figure supplement 4.** Peritoneal macrophages in aged mice do not show functional impairments.

**Figure supplement 4—source data 1.** Related to *Figure 1—figure supplement 4B–D*.

**Figure supplement 5.** Hematological parameters and erythropoietic activity in aging mice.

**Figure supplement 5—source data 1.** Related to *Figure 1—figure supplement 5A–H*.

**Figure supplement 6.** Gating strategy for erythroid progenitor cells in the spleen (**A**) and the bone marrow (**B**).

**Figure supplement 7.** Validation of the ferroportin antibody for flow cytometry.

**Figure supplement 7—source data 1.** Related to *Figure 1—figure supplement 7*.

**Figure supplement 8.** RPMs derived from aged IR *versus* aged mice show higher proliferation capacity.

**Figure supplement 8—source data 1.** Related to *Figure 1—figure supplement 8B*.

**Figure supplement 9.** RPM depletion in aged mice affects embryonically-derived RPMs and is not a consequence of diminished recruitment from monocytes.

**Figure supplement 9—source data 1.** Related to *Figure 1—figure supplement 9A–D*.

implies that the liver macrophage may partially compensate for the decreased EP activity in RPMs. In sum, we demonstrate that dietary iron restriction normalizes both systemic and RPM iron levels and reverses the age-related reduction in their numbers and iron-recycling ability.

## Aging triggers the retention of senescent RBCs, increased hemolysis, and the formation of non-degradable iron-rich aggregates in the spleen

Having established that dietary iron content in aging modulates RPM representation and EP capacity (*Figure 1L*), we examined parameters related to RBCs' fitness. First, we performed an RBC lifespan assay and found no differences in the rate of RBCs removal from the circulation between young and aged mice (*Figure 2A*, *Figure 2—figure supplement 1* for RBC gating strategy), suggesting intact fitness of circulating RBCs. Likewise, these RBCs did not show increased labile iron or ROS levels (*Figure 2B and C*). However, physiologically aged RBCs are first filtered in the spleen due to the loss of their elasticity, and this step is a prerequisite for their removal via EP (*Slusarczyk and Mleczko-Sanecka, 2021*). In agreement with this model, RBCs isolated from the spleen of young mice showed ROS build-up, a marker of their natural senescence (*Bratosin et al., 1998*; *Figure 2C*). We thus hypothesized that the reduced numbers and phagocytic capacity of RPMs in aging might impact local rather than systemic RBCs homeostasis. Consistently, splenic RBCs isolated from aged mice showed more pronounced ROS levels than RBCs derived from young mice (*Figure 2D*). In line with the partial mitigation of the defective iron-recycling capacity by restricted dietary iron content (*Figure 1K and L*), we found that mice fed an IR diet exhibited erythrocytic oxidative stress parameters similar to the young

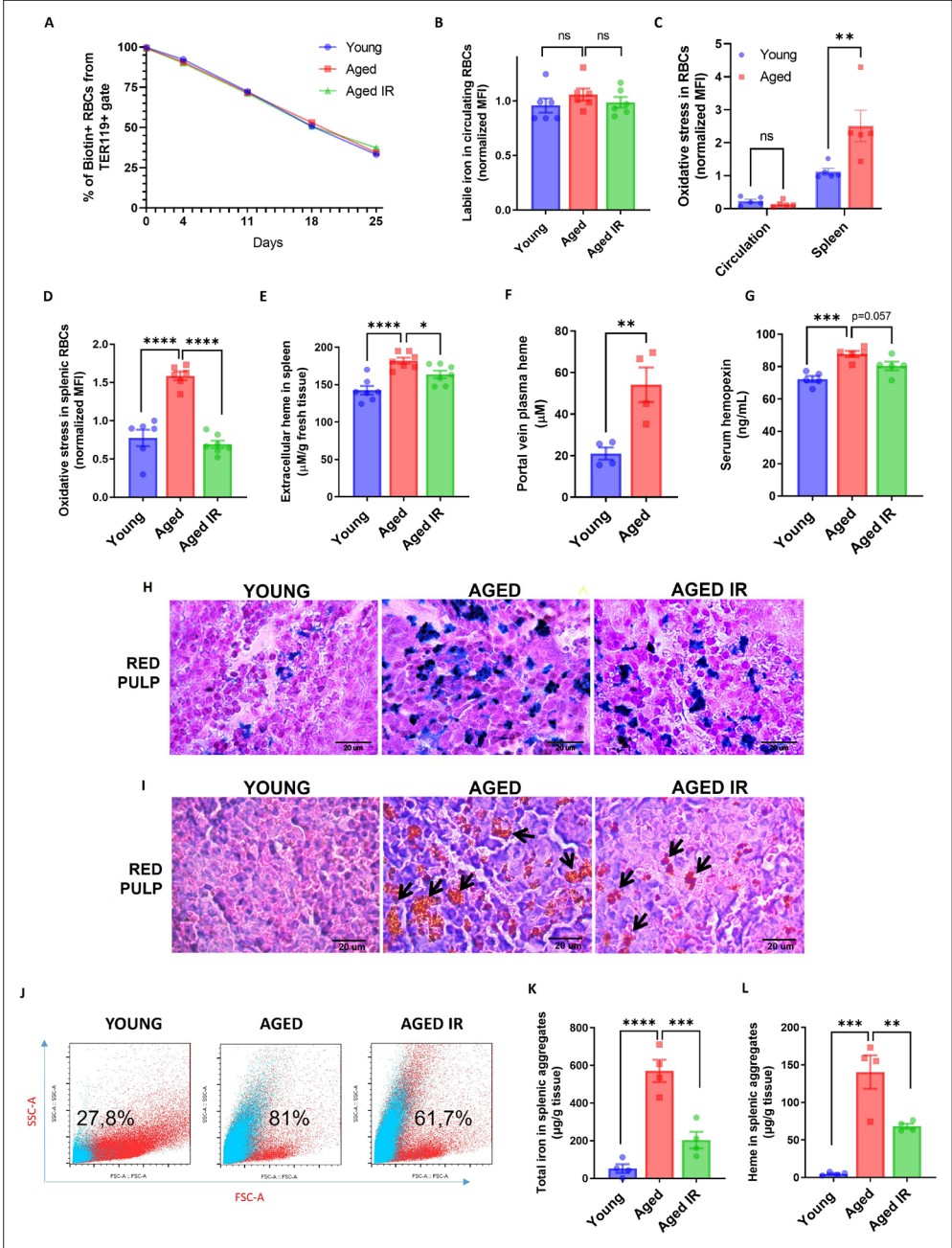

**Figure 2.** Aging triggers the retention of senescent RBCs, increased hemolysis, and the formation of non-degradable iron-rich aggregates in the spleen. (**A**) The RBC biotinylation lifespan assay was performed on circulating RBCs from young, aged, and aged IR mice. (**B**) Cytosolic ferrous iron (Fe2+) levels in RBCs derived from the circulation of young, aged, and aged IR mice were measured using FerroOrange with flow cytometry. (**C–D**) The cytosolic ROS levels in RBCs derived from (**C**) circulation and the spleen of young and aged mice; (**D**) the spleen of young, aged, and aged IR mice. Levels of ROS were estimated by determining CellROX Deep Red fluorescence intensity with flow cytometry. (**E**) Extracellular heme levels were measured in the supernatant obtained after the dissociation of spleens from young, aged, and aged IR mice using the Heme Assay Kit. (**F**) Portal vein plasma heme levels were measured in young, aged, and aged IR mice using Heme Assay Kit. (**G**) Serum hemopexin levels were measured in young, aged, and aged IR mice with Mouse Hemopexin ELISA Kit. (**H**) Perls' Prussian Blue staining of the splenic red pulp in young, aged, and aged IR mice. (**I**) Hematoxylin and eosin staining of the splenic red pulp in young, aged, and aged IR mice. Arrows indicate extracellular dark-colored aggregates. (**J**) Representative flow cytometry plots of magnetically-sorted splenocytes. In blue are events adverse for dead cell staining and cell surface markers CD45, TER119, CD11b, and F4/80. The percentage of these events in total

*Figure 2 continued on next page*

*Figure 2 continued*

acquired events is indicated. (**K**) The total iron content in magnetically-sorted, cell-free aggregates derived from spleens of young, aged, and aged IR mice was assessed using the Iron Assay Kit. (**L**) The heme content in magnetically-sorted, cell-free aggregates derived from spleens of young, aged, and aged IR mice was assessed using the Heme Assay Kit. Each dot in the graph (**A**) represents n=8. For the other panels, each dot represents one mouse. Data are represented as mean ± SEM. Welch's unpaired t-test determined statistical significance between the two groups; for the three groups, One-Way ANOVA with Dunnett's or Tukey's Multiple Comparison test was applied. In (**C**) Two-Way ANOVA test was applied. *p<0.05, **p<0.01, ***p<0.001 and ****p<0.0001.

The online version of this article includes the following source data and figure supplement(s) for figure 2:

**Source data 1.** Related to *Figure 2A–G and K–L*.

**Figure supplement 1.** Gating strategy for circulating (**A**) and splenic RBCs (**B**).

**Figure supplement 2.** FPN levels and iron status in splenic RBCs in aging.

**Figure supplement 2—source data 1.** Related to *Figure 2—figure supplement 2A–B*.

**Figure supplement 3.** Post-sorting purity of extracellular aggregates isolated from the aged spleen.

mice (*Figure 2D*). We further corroborated that the differences in ROS levels of splenic RBCs during aging reflect their senescence status, and are not a consequence of diminished FPN or LIP accumulation (*Figure 2—figure supplement 2A* and B). In agreement with the notion that a part of senescent RBCs undergoes local hemolysis (*Klei et al., 2020*), we further detected significantly higher levels of extracellular heme in the aged spleens, likely reflecting a higher burden of defective RBCs that evade efficient EP (*Figure 2E*). Interestingly, we found that the levels of heme in aging spleens were partially rescued by the IR diet. In support of these data, we detected higher levels of heme in plasma obtained from the portal vein (that carries the blood from the spleen to the liver) of aged *versus* young mice (*Figure 2F*) and higher serum concentrations of heme-scavenging hemopexin (*Figure 2G*). The latter tended to be rescued by the IR diet (p=0.057; *Figure 2G*). In conclusion, our data suggest that the reduced iron-recycling capacity of RPMs during aging may promote the retention of senescent RBCs prone to undergo local lysis and that these manifestations may be partially rescued by maintaining mice on an IR diet.

Previous work showed that genetic mouse models hallmarked by RPM depletion display splenic iron accumulation (*Haldar et al., 2014*; *Okreglicka et al., 2021*), partially resembling our observation in aged mice. However, the identity of splenic iron deposits in RPM-depleted spleens remains unclear. We thus aimed to characterize iron deposits in the aging spleen. Perls' Prussian blue staining that detects iron deposits showed that aged mice on a standard diet exhibited enhanced iron accumulation in the red pulp, a less pronounced phenotype in mice fed an IR diet (*Figure 2H*). This primarily reflected RPMs' iron status (*Figure 1I*). Intriguingly, eosin and hematoxylin staining visualized deposits in aged spleens that, in contrast to Perls' staining, were completely absent in young controls (*Figure 2I*). They appeared as large and extracellular in the histology sections and seemed smaller and less abundant in mice fed an IR diet (*Figure 2I*). Flow cytometry analyses confirmed that splenic single-cell suspension from aged but not young mice contained superparamagnetic particles, likely rich in iron oxide (*Franken et al., 2015*), that are characterized by high granularity as indicated by the SSC-A signal (*Figure 2J*). These particles did not show staining indicative of dead cells and failed to express typical splenocyte and erythroid markers (CD45, TER119, CD11b, or F4/80) (*Figure 2J*). With different granularity and size parameters, such particles appeared in mice aged on an IR diet, however, to a lesser extent. To further examine the composition of these aggregates, we established a strategy involving splenocyte separation in the lymphocyte separation medium followed by magnetic sorting, which successfully separated them from iron-rich cells (chiefly RPMs; *Figure 2—figure supplement 3*). We found that these aggregates contained large amounts of total iron (*Figure 2K*) and, interestingly, heme (*Figure 2L*), implying their partial RBC origin. Importantly, both iron and heme content was significantly reduced in mice fed an IR diet. In sum, our data show that aging is associated with local RBC dyshomeostasis in the spleen, and the formation of iron- and heme-rich aggregates, and that both these responses can be alleviated by limiting dietary iron content.

## Splenic age-triggered iron deposits are rich in aggregation-prone proteins derived from damaged RPMs

We next sought to explore the origin of the splenic iron aggregates. We imaged spleen sections from young and aged mice with transmission electron microscopy (TEM) to obtain ultrastructural information. We noticed that RPMs in young mice contained only intracellular dark-colored deposits, rich in structures that resemble ferritin (*Figure 3Ai*). Their appearance likely mirrors a form of iron deposits that were in the past referred to as hemosiderin, an intracellular clustered insoluble degradation product of ferritin (*Harrison and Arosio, 1996*; *Ward et al., 2000*). In older mice, we likewise observed intracellular deposits in morphologically intact cells (*Figure 3Aii*). However, in agreement with histological staining, we detected two other classes of aggregates: those that are still granular and enclosed within the membrane, but present in cells that are morphologically damaged (*Figure 3Aiii*) and those that are large, amorphic, and located extracellularly (*Figure 3Aiv*). Next, we conducted label-free proteomic profiling to determine the composition of magnetically-isolated aging-associated splenic aggregates. We identified over 3770 protein groups, among which 3290 were significantly more abundant in isolates from aged mice compared with the samples derived from young mice (*Figure 3B*). We assume that low-level detection of peptides from the young spleens may represent remnants from intracellular deposits that we observed via TEM and which may be released from RPMs during splenocyte processing. We performed a functional enrichment analysis of the top 387 hits (log2 fold change >5) with DAVID and ShinyGO bioinformatic tools (*Figure 3B and C*). We first noticed that although the identified proteins were derived from different organelles, one of the top characteristics that they shared was a disordered region in their amino acid sequence, a factor that may increase the risk of protein aggregation (*Uversky, 2009*). Consistently, the top hits were enriched in proteins related to neurodegeneration pathways, amyotrophic lateral sclerosis, Huntington's disease, and heat shock protein binding, linking their origin to proteostasis defects (*Medinas et al., 2017*). These included aggregation-prone huntingtin (HTT), amyloid-beta precursor protein (APP), and ataxin 3 (ATXN3) (*Lieberman et al., 2019*; *Liu et al., 2018* #251; *Figure 3B*). In agreement with the substantial iron load of the aggregates and the fact that iron promotes protein aggregation (*Klang et al., 2014*), we identified 'metal-binding' as another significant functional enrichment. We further observed that the top components of the aggregates included several proteins associated with immunoglobulin-like domains (*Figure 3B and C*), such as antibody chain fragments as well as Fc receptors, proteins associated with complement activation and phagocytosis, thus linking the aggregates to the removal of defective cells, likely senescent RBCs or damaged RPMs. Finally, consistent with our conclusions from EM imaging that cell death likely contributes to the formation of the aggregates (*Figure 3Aiv*), their top components were enriched with apoptosis-related protein (e.g. BID, DFFA, DAXX, and MRCH1; *Figure 3B*) and those linked to the response to stress (*Figure 3B and C*). Supporting the idea that iron-rich protein aggregates emerge primarily from damaged RPMs, and may likely contain some erythrocyte components, we observed an overlap between their protein components and the proteome of RPMs and erythrocytes (*Gautier et al., 2018*; *Figure 3D*).

To further explore the molecular mechanisms that influence their formation in aged spleens, we performed RNA sequencing of FACS-sorted RPMs from young and aged mice. We identified 570 differentially expressed genes, including 392 up- and 178 down-regulated transcripts (*Figure 3— figure supplement 1A*). RNA-seq data revealed rather an anti-inflammatory transcriptional pattern of aged RPMs, exemplified by the downregulation of genes encoding for MHC class II complexes and the induction of *Il10* (*Figure 3E* and *Figure 3—figure supplement 1A*). Most importantly, functional enrichment analysis showed that aged RPMs were hallmarked by ER stress, unfolded protein response, and ER-associated degradation (ERAD) (*Figure 3E*), regardless of the diet (*Figure 3—figure supplement 1B*). This observation corroborates the finding that aging-triggered splenic iron-rich protein deposits, enriched in aggregation prone-proteins, may result from proteotoxic stress in aged RPMs. ERAD is a pathway for targeting misfolded ER proteins for cytoplasmic proteasomal degradation (*Qi et al., 2017*). Correspondingly, we identified a significant increase in proteasomal activity in aged RPMs, measured with a dedicated fluorescent probe that undergoes proteasomal cleavage (*Figure 3F*). The IR diet reversed this proteasomal activation (*Figure 3F*). Since misfolded proteins can be also cleared from ER via ER-to-lysosome–associated degradation pathways (*De Leonibus et al., 2019*), and lysosomal activity was diminished in aged RPMs in a diet-dependent fashion, our data imply that proteasomal activity likely compensates for the reduced lysosomal-mediated protein quality

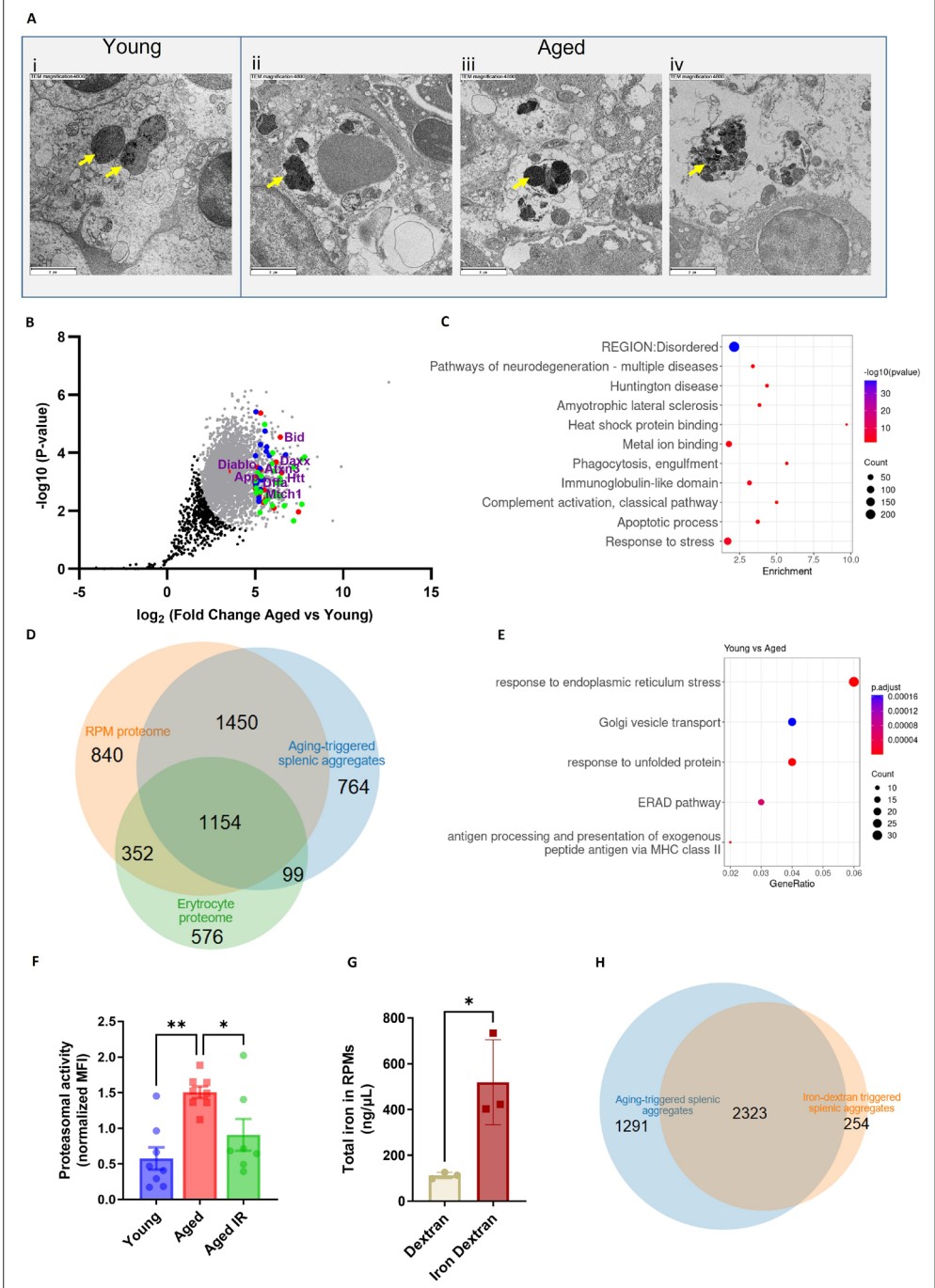

**Figure 3.** Splenic age-triggered iron deposits are rich in aggregation-prone proteins derived from damaged RPMs. (**A**) Ultrastructural analyses of the spleen red pulp sections by transmission electron microscopy. Arrows indicate dark-colored dense deposits. (**B**) Volcano plot illustrating 3290 protein groups (in gray) that are significantly more abundant in magnetically-sorted, cell-free aggregates derived from aged *versus* young spleen. Based on the functional enrichment analyses, the red color denotes proteins linked with 'pathways of neurodegeneration', green - those associated with an 'apoptotic process' and blue - those related to the 'immunoglobulin-like domain' category. (**C**) Enriched functional categories among the top 387 protein hits that are more abundant in magnetically-sorted, cell-free aggregates derived from aged *versus* young spleen. (**D**) Venn diagram illustrating the number of common proteins identified in RPMs, erythrocytes (human), and aging-triggered splenic aggregates. (**E**) Enriched functional categories among differentially regulated genes in FACS-sorted RPMs derived from young *versus* aged mice (identified by RNA-seq). (**F**) Proteasomal activity was measured in RPMs derived from

*Figure 3 continued on next page*

*Figure 3 continued*

young, aged, and aged IR mice using a fluorescent proteasome activity probe with flow cytometry. (**G**) The total intracellular iron content in magnetically sorted RPMs derived from spleens of dextran- and iron dextran-injected mice (8 hr post-injection) was assessed using the Iron Assay Kit. (**H**) Venn diagram illustrating the number of common protein groups identified in aging-triggered and iron dextran-triggered splenic aggregates (log2 fold change >1.5 *versus* respective young and dextran-injected controls, respectively; n=2). The dextran-triggered aggregates were isolated 24 hr post-injection. Each dot represents one mouse; in (B) three biological replicates per group were analyzed. Data are represented as mean ± SEM. Welch's unpaired t-test determined statistical significance between the two groups; statistical significance among the three groups was determined by One-Way ANOVA test with Tukey's Multiple Comparison test. *p<0.05, **p<0.01.

The online version of this article includes the following source data and figure supplement(s) for figure 3:

**Source data 1.** Related to *Figure 3B*.

**Source data 2.** Related to *Figure 3D*.

**Source data 3.** Related to *Figure 3F–G*.

**Source data 4.** Related to *Figure 3H*.

**Figure supplement 1.** Supplementary data related to RNA-seq analysis of RPMs derived from young, aged, and aged IR mice.

**Figure supplement 1—source data 1.** Related to *Figure 3—figure supplement 1A*.

**Figure supplement 2.** Components of lysosomes and ferritins are highly overrepresented in protein aggregates isolated from aged *versus* aged IR mice.

**Figure supplement 2—source data 1.** Related to *Figure 3—figure supplement 2A*.

---

control in RPMs during aging (*Figure 1M*). Finally, assigning a causal relationship between the iron loading of RPMs and their global proteostasis defect, we found that forced iron accumulation in RPMs in response to iron dextran injection in young mice (*Figure 3G*) provoked the formation of iron-rich protein aggregates whose composition largely overlaps with those from aged spleens (*Figure 3H*).

To explore how limited dietary iron content during aging affects the formation of splenic iron-rich deposits, we conducted TMT-based proteomic quantification of their composition. Out of 942 detected protein groups, 50 hits were significantly more abundant in magnetically-isolated aggregates from the aged mice compared to mice fed an IR diet (*Figure 3—figure supplement 2A*). Functional enrichment analysis identified components of lysosomes as by far the most overrepresented hits (*Figure 3—figure supplement 2A* and B), suggesting that alleviation of the lysosomal defects likely contributes to less pronounced aggregates formation in iron-reduced mice. The hits included also both H and L ferritin as being significantly increased in aggregates from aged mice on a standard diet. Of note, we failed to detect ferritin in the aggregates using antibody staining and flow cytometry, thus suggesting that cytoplasmic ferritin deficiency (*Figure 1G*) may result from ferritin aggregation.

Taken together, our data suggest that the splenic heme- and iron-rich deposits are formed of a broad spectrum of aggregation-prone protein debris that most likely originates from RPMs that have been damaged by proteotoxic stress.

## RPM dysfunction occurs early in aging and involves ferroportin downregulation

To explore early events that underlie RPM impairments during aging, we monitored in a time-wise manner how the formation of iron-rich aggregates corresponds to the functions of RPMs. We found that RPMs started to show increased LIP levels, and reduced their EP and lysosomal degradation capacity early during age progression (*Figure 4A–C*). Likewise, at 5 months of age, we observed that a larger proportion of splenic RBCs exhibited the senescence marker - increased ROS levels (*Figure 4D*). Interestingly, these changes reflected the accumulation of non-heme iron in the spleen and the appearance of splenic protein aggregates (*Figure 4E and F*). Since these deposits are insoluble, we assumed that their formation limits the bioavailability of iron for further systemic utilization. In support of this possibility, we noticed that the drop in transferrin saturation during aging coincided with RPM failure and the appearance of splenic aggregates (*Figure 4G*). To address the primary events that underlie RPM impairments we investigated mice very early during aging. We observed that ferroportin dropped significantly already in mice aged 4 months, which at least partially was

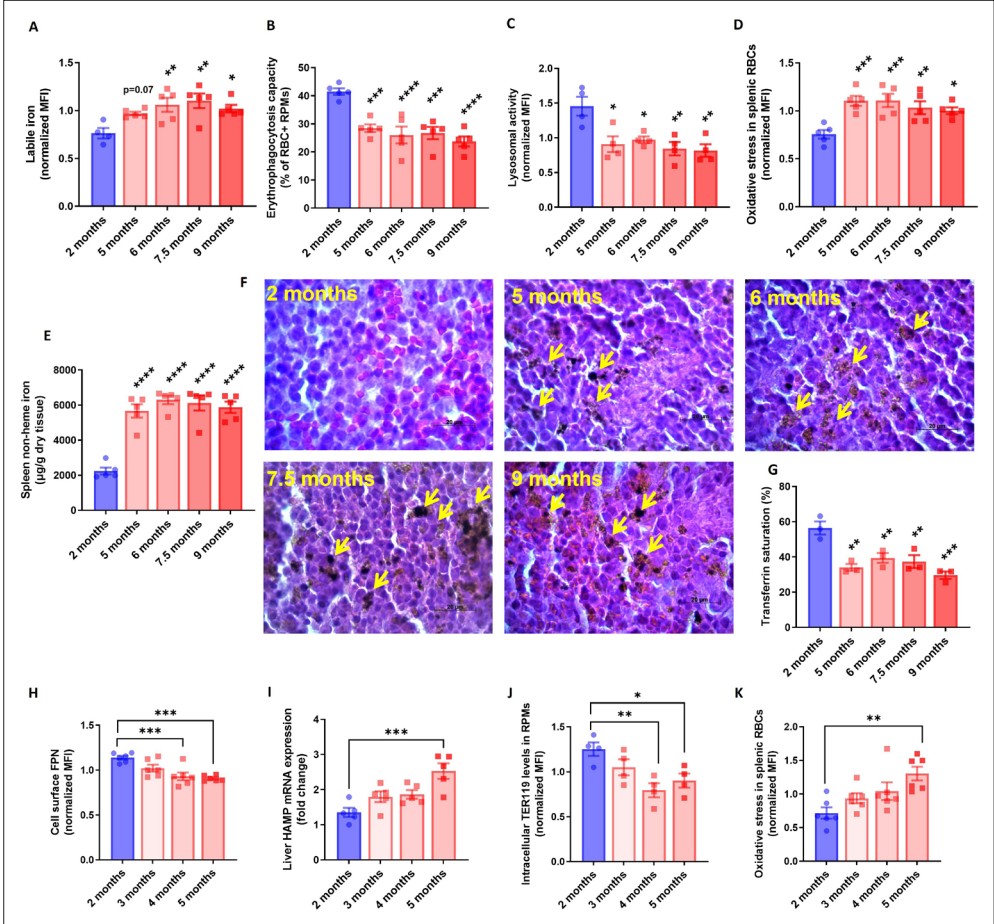

**Figure 4.** RPM dysfunction develops early in aging and is associated with ferroportin downregulation. (**A**) Cytosolic ferrous iron (Fe2+) content in RPMs derived from mice at the indicated age was measured using FerroOrange with flow cytometry. (**B**) Erythrophagocytosis capacity of RPMs derived from mice at the indicated age was determined using flow cytometry by measuring the percentage of RPMs that phagocytosed transfused PKH67-labeled temperature-stressed RBCs. (**C**) Lysosomal activity in RPMs derived from mice at the indicated age was determined using Lysosomal Intracellular Activity Assay Kit by flow cytometry. (**D**) The cytosolic ROS levels in RBCs derived from the spleens of mice at the indicated age were estimated by determining CellROX Deep Red fluorescence intensity with flow cytometry. (**E**) Splenic non-heme iron content was determined in mice at the indicated age. (**F**) Hematoxylin and eosin staining of the splenic red pulp in young, aged, and aged IR mice. Arrows indicate extracellular dark-colored aggregates. (**G**) Plasma transferrin saturation was determined in mice at the indicated age. (**H**) Expression of ferroportin (FPN) on the cell membrane of RPMs derived from mice at the indicated age was assessed by flow cytometry. (**I**) Relative mRNA expression of hepcidin (Hamp) in the liver from mice at the indicated age was determined by qPCR. (**J**) Erythrophagocytosis capacity of RPMs derived from mice at the indicated age was determined by intracellular staining of the erythrocytic marker TER119 in RPMs and flow cytometry. (**K**) The cytosolic ROS levels in RBCs derived from the spleens of mice at the indicated age were estimated by determining CellROX Deep Red fluorescence intensity with flow cytometry. Each dot represents one mouse. Data are represented as mean ± SEM. Statistical significance *versus* young controls (2 months) was determined by One-Way ANOVA test with Dunnett's Multiple Comparison test. *p<0.05, **p<0.01, ***p<0.001 and ****p<0.0001.

The online version of this article includes the following source data for figure 4:

**Source data 1.** Related to *Figure 4A–E and G–K*.

coupled with increased hepatic hepcidin expression (*Figure 4H, I*). At the same time point, the cells started to exhibit decreased EP activity [here determined by intracellular staining of the erythrocytic marker TER119 in RPMs *Akilesh et al., 2019*; *Figure 4J*], and only afterward, at the age of 5 months, we detected significant retention of senescent, ROS-rich RBCs in the spleen (*Figure 4K*). In sum, our

data show that iron-triggered dysfunction of RPMs is initiated early during aging, likely due to ferroportin downregulation.

## Aging triggers RPM loss via mechanisms that resemble ferroptosis and involve proteotoxicity

Increased iron burden in RPMs was shown previously to drive ferroptotic cell death upon acute transfusion of damaged RBCs (*Youssef et al., 2018*). Since RPM representation was decreased in aged spleens (*Figure 1K*) and the TEM imaging revealed damaged iron-loaded RPMs (*Figure 3A*), we speculated that RPMs may undergo spontaneous ferroptosis during aging. In agreement with this hypothesis, both labile iron (*Figure 4A*), a factor that promotes ferroptosis, and lipid peroxidation (*Figure 5A*), a ferroptosis marker (*Dixon et al., 2012*; *Friedmann Angeli et al., 2014*) increased in RPMs during aging. Importantly, we observed that feeding mice an IR diet during aging diminished lipid peroxidation to the levels characteristic of young mice (*Figure 5B*), similar to what we observed for the labile iron build-up (*Figure 1H*). In line with these findings, we observed that injection of iron dextran in young mice led to robust lipid peroxidation in RPMs (*Figure 5C*) and their mild depletion (p=0.056; *Figure 5D*) at 8 hr post-injection, before the formation of splenic protein aggregates occurred (*Figure 3H*).

Earlier studies reported that ferroptotic cells are characterized by reduced mitochondria size and diminished mitochondrial membrane potential but not mitochondrial oxidative stress (*Chen et al., 2021*; *Dixon et al., 2012*; *Friedmann Angeli et al., 2014*; *Neitemeier et al., 2017*). Consistent with these data, we failed to detect augmented ROS levels in the mitochondria of aged RPMs (*Figure 5— figure supplement 1A*). Instead, we observed that mitochondria mass (*Figure 5E*), as well as mitochondrial activity (*Figure 5F*), decreased in RPMs of aged mice, and both these phenotypes were alleviated by an IR diet. This corresponded to higher ATP levels in FACS-sorted RPMs derived from aged mice fed an IR diet *versus* a standard diet (*Figure 5—figure supplement 1B*). Previous ultrastructural studies showed that mitochondria of ferroptotic cells are hallmarked by reduced size and fewer cristae, increased membrane density, and some appear swollen (*Chen et al., 2021*; *Dixon et al., 2012*; *Friedmann Angeli et al., 2014*). Consistently, our TEM imaging showed that in contrast to the RPMs of young mice, those from aged animals exhibited the above mitochondrial defects with some appearing disintegrated (*Figure 5G*). Mitochondria from mice fed an IR diet were small and dense, but they displayed a lesser degree of swelling and damage.

Since aged RPMs are characterized by proteotoxic stress and iron retention, we investigated how these two factors contribute to their viability. To this end, we employed a cellular model of RPMs [which we termed induced (i)-RPMs] that were generated by exposing bone marrow-derived macrophages to hemin and IL-33, two factors that drive RPM differentiation (*Haldar et al., 2014*; *Lu et al., 2020*) and induce RPM-like transcriptional signatures (*Lu et al., 2020*). We observed that iron loading (with ferric ammonium citrate [FAC]) and blockage of ferroportin by synthetic mini-hepcidin (PR73) (*Stefanova et al., 2018*) caused protein aggregation in iRPMs and led to their decreased viability but only in cells that were exposed to heat shock, a well-established trigger of proteotoxicity (*Figure 5H, I*). Interestingly, we observed that although FAC treatment increased lipid peroxidation to a similar extent in intact cells and those exposed to heat shock (*Figure 5J*), cell viability was only reduced when the iron loading and proteotoxic challenge were combined (*Figure 5K*). Even more strikingly, the latter synergistic cytotoxic effect was prevented by the ferroptosis blocker Liproxstatin-1 (*Friedmann Angeli et al., 2014*; *Figure 5K*). Finally, consistent with the observation that apoptosis markers can be elevated in tissues upon ferroptosis induction (*Friedmann Angeli et al., 2014*) and proteotoxicity may lead to apoptosis (*Brancolini and Iuliano, 2020*), we detected DNA fragmentation (via TUNEL assay) in the spleens of aged mice, a phenotype that was less pronounced in mice on an IR diet (*Figure 5— figure supplement 2*). Taken together, our data suggest that RPM demise during aging is driven by iron-dependent cell death with ferroptosis characteristics coupled with proteostasis collapse.

## Iron loading but not oxidative stress undermines the phagocytic activity of aged RPMs in concert with impaired heme metabolism and ER stress

The contribution of iron deposition to age-related deterioration of cellular functions is believed to be primarily mediated by the pro-oxidative properties of labile iron (*Xu et al., 2008*). We observed

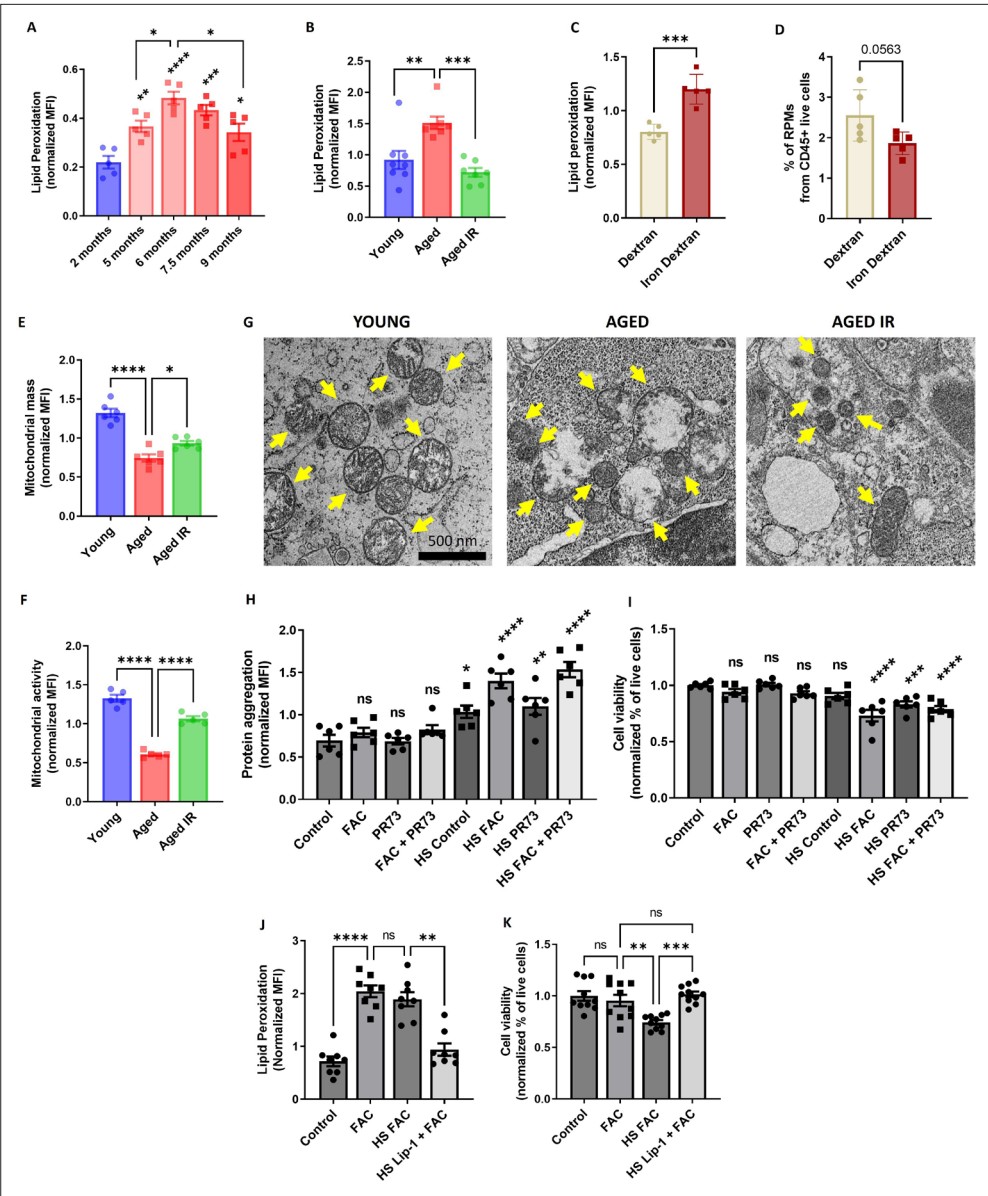

**Figure 5.** Aging triggers RPM loss via mechanisms that resemble ferroptosis and involve proteotoxicity. (**A**) Lipid peroxidation was determined in RPMs derived from mice at the indicated age using the Lipid Peroxidation Assay Kit with flow cytometry. (**B**) Lipid peroxidation was determined in RPMs derived from young, aged, and aged IR mice using the Lipid Peroxidation Assay Kit with flow cytometry. (**C**) Lipid peroxidation was determined in RPMs derived from dextran- and iron-dextran-injected mice (8 hr post-injection) using the Lipid Peroxidation Assay Kit with flow cytometry. (**D**) The percentage of RPMs from CD45+ live cells present in the spleen of dextran- and iron dextran-injected mice (8 hr post-injection) was assessed by flow cytometry. (**E**) Mitochondrial mass and (**F**) mitochondrial activity were determined in RPMs derived from young, aged, and aged IR mice using MitoTracker Green and TMRE probes, respectively, with flow cytometry. (**G**) Ultrastructural analyses of mitochondrial morphology in spleen red pulp sections obtained from young, aged, and aged IR mice. Yellow arrows indicate mitochondria. (**H**) Protein aggregation and (**I**) cell viability in cultured iRPMs were determined using PROTEOSTAT Aggresome detection kit and a fluorescent Aqua Live/Dead probe, respectively, with flow cytometry. Cells were treated with FAC (150 µM, 24 hr), PR73 mini-hepcidin (2 µg/mL, 24 hr), or exposed to heat shock (HS) stress (42 °C, 4 hr) as indicated. (**J**) Lipid peroxidation and (**K**) cell viability of cultured iRPMs were determined using the Lipid Peroxidation Assay Kit and fluorescent Aqua Live/Dead probe, respectively, with flow cytometry. Cells were treated with FAC (150 µM, 24 hr), Liproxstatin-1 (Lip-1; 2 µM, 25 hr) or exposed to heat shock (HS) stress (42 °C, 4 hr) as indicated. Each dot represents one mouse or independent cell-based experiment. Data are represented as mean ± SEM. Welch's unpaired t-test determined statistical significance between the two groups; statistical significance

*Figure 5 continued on next page*

*Figure 5 continued*

among the three or more groups was determined by One-Way ANOVA test with Dunnett's or Tukey's Multiple Comparison test. *p<0.05, **p<0.01, ***p<0.001 and ****p<0.0001.

The online version of this article includes the following source data and figure supplement(s) for figure 5:

**Source data 1.** Related to *Figure 5A–F and H–K*.

**Figure supplement 1.** Mitochondrial oxidative stress and ATP levels in young, aged and aged IR RPMs.

**Figure supplement 1—source data 1.** Related to *Figure 5—figure supplement 1A–B*.

**Figure supplement 2.** Apoptotic cell death was detected using In Situ Cell Death Detection Kit in spleen sections of young, aged and aged IR mice spleens.

a clear indication of oxidative stress in aging RPMs (*Figure 1J*) that was rescued by an IR diet. Next, we aimed to verify whether the increased ROS levels solely contribute to RPM decline during aging. To this end, we supplemented aging mice with the antioxidant N-Acetyl-L-cysteine (NAC), which was previously shown to revert aging-related physiological changes (*Berman et al., 2011*; *Ma et al., 2016*). With this strategy, we successfully reduced oxidative stress in aged RPMs (*Figure 6—figure supplement 1A*). However, we did not observe an improvement in the EP capacity of RPMs isolated from NAC-supplemented mice compared to untreated aged mice (*Figure 6—figure supplement 1B*). Consistently, the retention of senescent RBCs in the spleen (*Figure 6—figure supplement 1C*) and their local hemolysis (*Figure 6—figure supplement 1D*) were not rescued by the NAC administration. Likewise, enhanced RPM lipid peroxidation (*Figure 6—figure supplement 1E*), a ferroptosis marker, the formation of insoluble aggregates (*Figure 6—figure supplement 1F*), and splenic iron overload (*Figure 6—figure supplement 1G*) were equally present in aged mice regardless of the NAC treatment. These data suggest that RPM dysfunction and damage during aging are chiefly driven by RPM iron loading rather than excessive ROS.

Next, we investigated in more detail the causative role of iron loading in suppressing EP. We confirmed that iRPMs significantly decreased the ability for RBC uptake upon iron overload with FAC in a manner that was partially rescued by the iron chelator desferrioxamine (DFO; *Figure 6A and B*). In addition, we observed that iron loading by FAC suppressed lysosomal activity, an important factor for RBC degradation (*Figure 6C*). We also found that mitochondria membrane potential was similarly diminished by FAC and DFO alone, thus uncoupling the phagocytic and lysosomal activity of iRPMs from their mitochondria fitness (*Figure 6D*). Mechanistically, we found that iron loading reduced protein expression levels of the receptors that recognize phosphatidylserine on apoptotic cells MERTK, AXL, and TIM4, and which are highly expressed by RPMs (*Slusarczyk and Mleczko-Sanecka, 2021*), likely contributing to EP (*Figure 6E–G*). Of note, expression levels of another phosphatidylserine receptor STAB2 or the Fc receptor CD16 that as well are present in RPMs (*Slusarczyk and Mleczko-Sanecka, 2021*) were unchanged upon iron/DFO treatment of iRPMs (*Figure 6—figure supplement 2*). Consistent with the important role of calcium signaling in the regulation of EP (*Ma et al., 2021*), we observed that iron accumulation reduced calcium levels in iRPMs (*Figure 6H*). Importantly, the effects of iron excess on the levels of the apoptotic cell receptors and calcium were rescued by DFO.

Interestingly, our RNA-seq data revealed that one of the transcripts that significantly decrease in aged RPMs is *Hmox1* (*Figure 3—figure supplement 1A*), which encodes HO-1. We validated this result using intracellular staining and flow cytometry, and uncovered a marked decrease in HO-1 protein level in RPMs during aging (*Figure 6I and J*; see *Figure 6—figure supplement 3* for antibody validation). Correspondingly, the blockage of HO-1 in iRPMs with zinc protoporphyrin (ZnPP) suppressed the EP intensity (*Figure 6K*) and, to a lesser extent, the capacity for lysosomal degradation (*Figure 6—figure supplement 4A*). Next, we explored why ZnPP, but not hemin alone (*Figure 6—figure supplement 4B*), suppressed EP. We speculated that possibly another co-product of HO-1 enzymatic activity (CO or/and biliverdin), which we expect to be high in hemin-exposed iRPMs, counterbalances the suppressive effect of iron release after EP. Interestingly, we found that the CO donor CORM-A1, but not biliverdin, rescued EP capacity under ZnPP exposure in iRPMs (*Figure 6—figure*

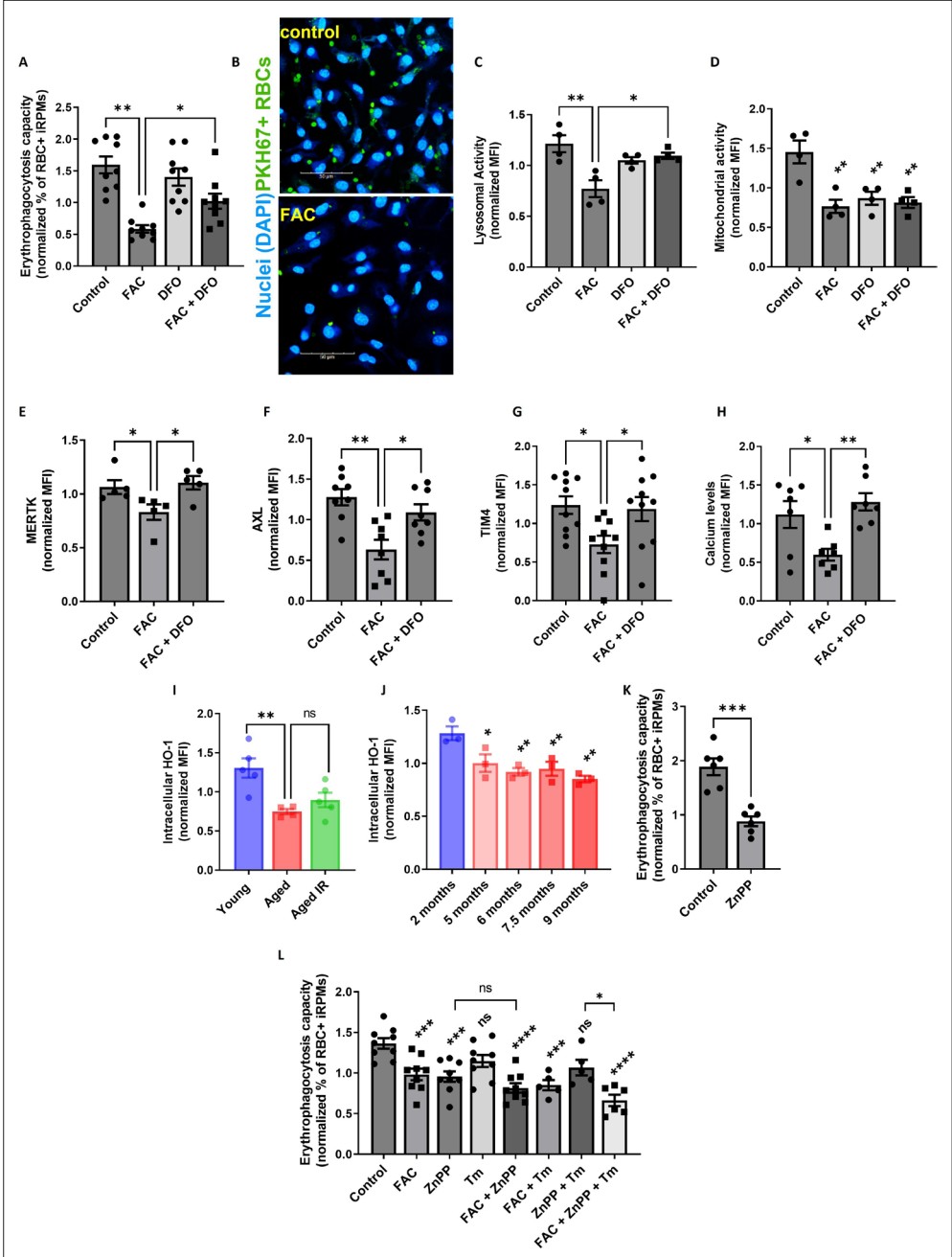

**Figure 6.** Iron loading undermines the phagocytic activity of iRPMs in concert with impaired heme metabolism and ER stress. (**A**) Normalized erythrophagocytosis capacity of PKH67-labeled temperature-stressed RBCs by cultured iRPMs. Cells were treated with FAC (50 µM, 24 hr) and DFO (100 µM, 18 hr) as indicated. (**B**) Representative confocal microscopy images of erythrophagocytosis in FAC-treated iRPMs compared with control cells. (**C**) Lysosomal and (**D**) mitochondrial activity of cultured iRPMs were determined using Lysosomal Intracellular Activity Assay Kit and TMRE probe, respectively, with flow cytometry. Cells were treated with FAC (50 µM, 24 hr) and DFO (100 µM, 18 hr) as indicated. (**E**) Cell membrane expression levels of MERTK, (**F**) AXL and (**G**) TIM4 of cultured iRPMs were determined by using fluorescently labeled antibodies and flow cytometry. Cells were treated with FAC (50 µM, 24 hr) and DFO (100 µM, 18 hr) as indicated. (**H**) Cytosolic calcium levels of cultured iRPMs were determined using Cal-520 fluorescent probe with flow cytometry. Cells were treated with FAC (50 µM, 24 hr) and DFO (100 µM, 18 hr) as indicated. (**I**) Intracellular HO-1 protein levels in RPMs isolated from young, aged, and aged IR mice were measured by flow cytometry. (**J**) Intracellular HO-1 protein levels in RPMs isolated from mice at the indicated age were measured by flow cytometry. (**K**) Normalized erythrophagocytosis capacity of PKH67-labeled temperature-stressed RBCs by cultured iRPMs. Cells were treated with ZnPP (0.5 µM, 24 hr). (**L**) Normalized

*Figure 6 continued on next page*

*Figure 6 continued*

erythrophagocytosis capacity of PKH67-labeled temperature-stressed RBCs by cultured iRPMs. Cells were treated with indicated concentrations of ZnPP (0.5 µM) and FAC (10 µM) and with ER stress inducer Tunicamycin (Tm; 2.5 µM) for 24 hr. Each dot represents one mouse or independent cell-based experiment. Data are represented as mean ± SEM. Statistical significance among the three or more groups was determined by One-Way ANOVA test with Dunnett's or Tukey's Multiple Comparison test. *$p<0.05$, **$p<0.01$, ***$p<0.001$ and ****$p<0.0001$.

The online version of this article includes the following source data and figure supplement(s) for figure 6:

**Source data 1.** Related to *Figure 6A and C–L*.

**Figure supplement 1.** Iron loading, but not oxidative stress leads to RPM decline during aging.

**Figure supplement 1—source data 1.** Related to *Figure 6—figure supplement 1A–E and G*.

**Figure supplement 2.** The Fc receptor CD16 and the apoptotic cell receptor STAB2 are not regulated by iron-loading.

**Figure supplement 2—source data 1.** Related to *Figure 6—figure supplement 2A–B*.

**Figure supplement 3.** Validation of the antibody against HO-1 in flow cytometry.

**Figure supplement 3—source data 1.** Related to *Figure 6—figure supplement 3*.

**Figure supplement 4.** The effects of HO-1 inhibition on lysosomal activity and HO-1 co-products on the erythrophagocytosis capacity of iRPMs.

**Figure supplement 4—source data 1.** Related to *Figure 6—figure supplement 4A–D*.

---

*supplement 4C* and D), suggesting that CO acts as a modulator of EP activity in RPMs. Whether deficient CO levels in RPMs *in vivo* contribute to EP suppression would require further investigation. However, in support of this possibility, our RNA-seq data implied that enzymes involved in the pentose phosphate pathway, such as *Tkt*, a transcriptional target of CO-mediated regulation (*Bories et al., 2020*), are downregulated in aged RPMs (*Figure 3—figure supplement 1A*).

RPMs derived from aging mice show similar HO-1 levels (*Figure 6I*) and the degree of ER stress (*Figure 3—figure supplement 1B*) irrespective of diet and differ primarily in iron status (*Figure 1H, I*). Therefore, we next tested how the combination of ZnPP and the ER stress inducer tunicamycin with a mild FAC exposure (10 µM) affects EP. Although ER stress alone did not significantly affect EP, we uncovered that FAC treatment elicited an additive effect with combined exposure to ZnPP and tunicamycin, but not with ZnPP alone (*Figure 6L*). Taken together, our data imply that iron content in RPMs represents the key factor that determines their RBC clearance potential, and its excess may exacerbate the suppressive impact of HO-1 loss and ER stress on erythrophagocytosis.

## Discussion

RPMs play an essential role in preventing the release of hemoglobin from RBCs and ensuring the turnover of the body an iron pool. To date, impairments of iron recycling were reported in genetically modified mice, hallmarked by the loss of RPMs due to the absence of genes that control their differentiation (*Kohyama et al., 2009*; *Okreglicka et al., 2021*). Here, for the first time, we provide evidence that the RPM damage and defective capacity for RBC clearance accompany physiological aging (*Figure 7*).

The present study provides new evidence for the origin and nature of iron deposits in the aging spleen. Recent findings show that aging promotes a global decrease in protein solubility (*Sui et al., 2022*). The longevity of embryonically-derived RPMs, reaching at least 32 weeks in mice (*Hashimoto et al., 2013*; *Liu et al., 2019*), puts pressure on these cells to handle the constant flux of iron, a factor that increases protein insolubility (*Klang et al., 2014*). This may render RPMs particularly sensitive to the build-up of protein aggregates. Consistently, we found a clear transcriptional signature for unfolded protein response and ER stress in aged RPMs, which may be linked to the constitutive phagocytic activity of this macrophage population (*Kim et al., 2018*). We propose that unrestricted iron availability from the diet further ameliorates RPM fitness during aging, chiefly via early ferroportin downregulation and intracellular iron accumulation. The latter further decreases protein solubility and

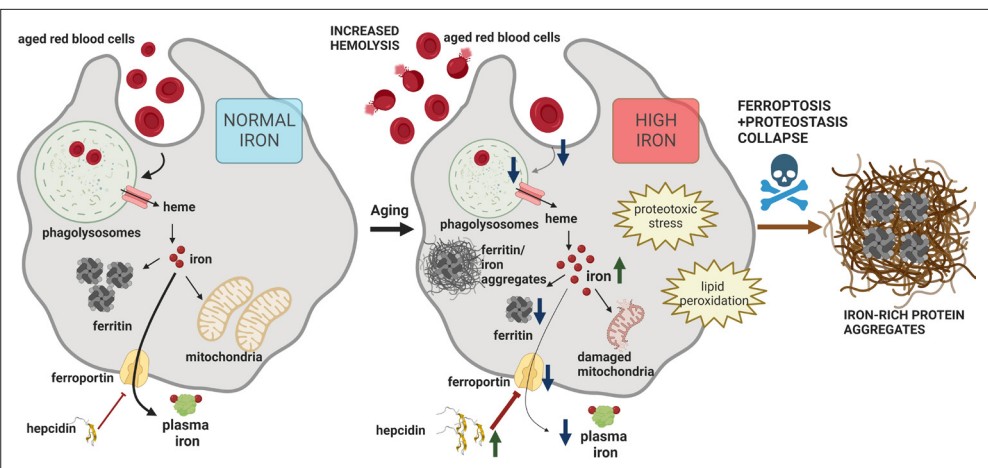

**Figure 7.** Model of RPM dysfunction and collapse during aging. Iron-dependent functional defects of RPMs are initiated early during aging. Intracellular iron loading of RPMs, in concert with proteostasis defects, precedes the deposition of iron in a form of extracellular protein-rich aggregates, likely emerging from damaged RPMs. The build-up of un-degradable iron-rich deposits, in concert with increased hepcidin levels, and reduced erythrophagocytic and lysosomal activity of the remaining RPMs limit plasma iron availability during aging. A drop in RBC clearance capacity leads to enhanced splenic hemolysis.

inhibits lysosomal activity which is critical for autophagy-mediated protein quality control (*Kaushik and Cuervo, 2015*). Critically, an increase in labile iron ultimately leads to the appearance of toxic lipid peroxides, the oxidation products that drive ferroptosis (*Dixon et al., 2012*). Our results show that only the combination of proteotoxic stress, likely associated *in vivo* with specialized iron-recycling functions of RPMs, with iron loading or ferroportin blockage, is cytotoxic to the cellular model of RPMs. The fact that Liproxstatin-1, a ferroptosis inhibitor (*Friedmann Angeli et al., 2014*), rescues cell viability in this context implies the involvement of ferroptosis, but how mechanistically proteome instability and iron-triggered toxicity synergize remains an open question.

Thus far ferroptotic cell death was mainly described in genetic models, for example with disturbed detoxification of lipid peroxides due to conditional loss of *Gpx4* (*Friedmann Angeli et al., 2014*; *Matsushita et al., 2015*) or in various pathologies (*Park et al., 2019*; *Wang et al., 2019*; *Yan et al., 2021*). Here, we propose that ferroptosis may critically contribute to RPM loss during physiological aging. Notably, the excessive clearance of damaged RBCs in mouse models of blood transfusion, hemolytic anemia, or the anemia of inflammation induced by heat-killed *Brucella abortus* was previously shown to cause transient or sustained RPMs depletion (*Haldar et al., 2014*; *Theurl et al., 2016*; *Youssef et al., 2018*). Although the splenic niche may be replenished to some extent via splenic monocyte recruitment and/or resident RPMs proliferation (*Haldar et al., 2014*; *Youssef et al., 2018*), our data imply that the damage of embryonically derived RPM in these models may lead to the irreversible formation of insoluble iron-rich splenic protein aggregates. Premature clearance of defective RBCs also hallmarks β-thalassemia (*Slusarczyk and Mleczko-Sanecka, 2021*). To the best of our knowledge, RPM loss was not extensively studied in a mouse model of thalassemia. However, interestingly, thalassemic mice display enhanced splenic Perls'-stained deposits that appear to be extracellular and are not amenable to iron chelation therapy (*Sanyear et al., 2020*; *Vadolas et al., 2021*), possibly resembling those identified by the present study. Of note, genetic disruption of *Hrg1* leads to the formation of heme aggregate hemozoin inside enlarged phagolysosomes (*Pek et al., 2019*). The large size, extracellular localization, and high content of proteins distinguish age-triggered splenic aggregates from hemozoin. However, we cannot exclude the possibility that this polymer-like structure may expand during aging and contribute to the formation of splenic iron deposits.

Our study identified three new factors that affect the intensity of EP, the iron content, the activity of HO-1, and, to less extent, ER stress. Among them, iron accumulation in RPMs, independently of ROS generation, emerged as a major driver of EP suppression during aging, likely via the downregulation of apoptotic cell receptors (*Slusarczyk and Mleczko-Sanecka, 2021*) and calcium signaling (*Ma et al., 2021*). The underlying mechanisms for excessive iron deposition in aged RPMs most likely include reduced FPN protein levels and may be further aggravated by the sequestration of ferritin in protein aggregates via the mechanisms that remain to be explored in detail. It is also plausible that, like neurons, aged RPMs show low capacity for heme synthesis (*Atamna et al., 2002*), which may lead to defective iron utilization in the mitochondria and increased cytoplasmic iron burden. Mitochondrial damage that we identified in aged RPMs may support this possibility. We also demonstrated that iron loading decreases the capacity of RPMs for phagolysosomal RBC degradation. Of note, patients who suffer from lysosomal storage disorder, Gaucher disease, show iron deposits in the splenic macrophages (*Clarke et al., 2021*; *Lefebvre et al., 2018*). Iron sequestration in Gaucher macrophages was attributed to the hepcidin-mediated downregulation of ferroportin, or more recently to increased erythrophagocytosis of Gaucher RBCs (*Dupuis et al., 2022*). Based on our study, it is plausible that defects in proteostasis, due to defective lysosomal-mediated protein quality control, may contribute to aberrant iron management in Gaucher macrophages.

HO-1 is well characterized for its cytoprotective, anti-oxidative, and anti-inflammatory functions, (*Gozzelino et al., 2010*), and its activity was proposed to prevent cellular senescence (*Even et al., 2018*; *Hedblom et al., 2019*; *Luo et al., 2018*; *Suliman et al., 2017*). Since RPMs are key cells where HO-1 exerts its enzymatic function (*Vijayan et al., 2018*), our observation that HO-1 levels decrease in these cells during aging is of high physiological significance. Although aged RPMs show some hallmarks of senescence linked to HO-1 deficiency, such as defective mitochondria or excessive ROS, they do not exhibit proinflammatory gene expression signatures. The suppression of EP by inhibition of HO-1 activity may be mediated by heme overload, as shown previously for general phagocytosis (*Martins et al., 2016*). However, we provide evidence that another product of HO-1, CO, restores the EP capacity of cells subjected to HO-1 blockage, thus emerging as an inducer of RBC uptake. These findings are in agreement with the previous work, reporting that CO administration protects *Hmox1* knock-out mice from sepsis-induced lethality via stimulation of bacterial phagocytosis (*Chung et al., 2008*).

Collectively, our study implies that intracellular iron loading of RPMs impairs their EP capacity and synergistically with proteostasis defects leads to their demise. These impairments result in local RBC dyshomeostasis in the spleen, due to the shift from phagocytosis to splenic RBC lysis (*Klei et al., 2020*), and the formation of extracellular proteinaceous aggregates, rich in iron and heme, emerging from damaged RPMs (*Figure 7*). We propose that the build-up of un-degradable iron-rich particles, in concert with increased hepcidin levels, and limited iron-recycling activity of the remaining RPMs limit plasma iron availability during aging. Future studies may address whether the consequences of RPM dysfunction for the splenic microenvironment, together with their anti-inflammatory transcriptional profile, contribute to the immunosenescence that accompanies aging (*Nikolich-Žugich, 2018*). Interestingly, microglia iron overload and ferroptosis were recently identified as critical drivers of neurodegenerative diseases (*Kenkhuis et al., 2021*; *Ryan et al., 2023*). In such context, our findings suggest that in addition to various iron chelation approaches (*Liu et al., 2018*), long-term reduction of dietary iron intake may represent an alternative for neurodegeneration prevention. Finally, our study has some limitations. We characterized in detail RPM dysfunction in female mice on a relatively low-iron C57BL/6 J background. Although we have observed similar phenotypes in Balb/c females, we failed to observe the formation of splenic iron-rich protein aggregates in C57BL/6 J males (data not shown). Having established that the degree of RPM collapse depends on dietary iron and internal hepcidin-ferroportin regulatory circuitry, it remains to be investigated how it is manifested in male mice on iron-rich genetic backgrounds or humans. Lastly, since standard mouse diets contain more-than-sufficient amounts of iron, our study could be also interpreted as the effect of long-term exposure to excessive iron on RPM functions.

# Materials and methods

**Key resources table**

| Reagent type (species) or resource | Designation | Source or reference | Identifiers | Additional information |
| --- | --- | --- | --- | --- |
| Antibody | Anti-mouse CD45 APC/Cyanine7 (Rat monoclonal) | BioLegend | cat.# 103115 | 1:200 |
| Antibody | Anti-mouse/human CD45R/B220 Pacific Blue (Rat monoclonal) | BioLegend | cat.# 103230 | 1:200 |
| Antibody | Anti-mouse F4/80 APC (Rat monoclonal) | BioLegend | cat.# 123115 | 1:100 |
| Antibody | Anti-mouse F4/80 FITC (Rat monoclonal) | BioLegend | cat.# 123107 | 1:100 |
| Antibody | Anti-mouse/human CD11b PE/Cyanine7 (Rat monoclonal) | BioLegend | cat.# 101215 | 1:200 |
| Antibody | Anti-mouse Treml4 PE (Rat monoclonal) | BioLegend | cat.# 143303 | 1:100 |
| Antibody | Anti-mouse TER-119 Pacifin Blue (Rat monoclonal) | BioLegend | cat.# 116231 | 1:300 |
| Antibody | Anti-mouse TER-119 FITC (Rat monoclonal) | BioLegend | cat.# 116205 | 1:300 |
| Antibody | Anti-mouse CD71 APC (Rat monoclonal) | BioLegend | cat.# 113819 | 1:100 |
| Antibody | Anti-mouse CD71 PE (Rat monoclonal) | BioLegend | cat.# 113807 | 1:100 |
| Antibody | Anti-mouse TER-119 PE (Rat monoclonal) | BioLegend | cat.# 116207 | 1:300 |
| Antibody | Anti-mouse CD45 PE/Cyanine7 (Rat monoclonal) | BioLegend | cat.# 147703 | 1:200 |
| Antibody | Anti-mouse FPN (Rat monoclonal) | Amgen | DOI:10.1172/JCI127341 | 1:100 |
| Antibody | Anti-mouse Gr-1 Pacific Blue (Rat monoclonal) | BioLegend | cat.# 108429 | 1:200 |
| Antibody | Anti-mouse MHCII Pacific Blue (Rat monoclonal) | BioLegend | cat.# 107619 | 1:200 |
| Antibody | Anti-human/mouse Ferritin Heavy Chain (FTH1) (Rabbit polyclonal) | Cell Signaling Technology | cat.# 3998 | 1:100 |
| Antibody | Anti-mouse AXL APC (Rat monoclonal) | Invitrogen | cat.# 17-1084-82 | 1:100 |
| Antibody | Anti-mouse MERTK FITC (Rat monoclonal) | BioLegend | cat.# 151503 | 1:100 |
| Antibody | Anti-mouse Tim-4 Alexa Fluor 647 (Rat monoclonal) | BioLegend | cat.# 130007 | 1:100 |
| Antibody | Anti-mouse Stab2 Alexa Fluor 488 (Rabbit polyclonal) | G-Biosciences | cat.# ITN2255 | 1:100 |
| Antibody | Anti-mouse/human Ferritin Light Chain (FTL) (Rabbit polyclonal) | Abcam | cat.# ab69090 | 1:100 |
| Antibody | Anti-mouse HO-1 (Rabbit polyclonal) | Enzo Life Sciences | cat.# ADI-OSA-150 | 1:400 |

*Continued on next page*

*Continued*

| Reagent type (species) or resource | Designation | Source or reference | Identifiers | Additional information |
|---|---|---|---|---|
| Antibody | Anti-mouse Ki-67 Alexa Fluor 488 (Rat monoclonal) | BioLegend | cat.# 652417 | 1:100 |
| Antibody | Anti-Rabbit IgG (H+L) Alexa Fluor 488 (Donkey polyclonal) | Thermo Fisher Scientific | cat.# A-21206 | 1:200 |
| Antibody | Anti-Rabbit IgG (H+L) Alexa Fluor 647 (Donkey polyclonal) | ThermoFisher Scientific | cat.# A-31573 | 1:200 |
| Antibody | Anti-mouse CD16/32 TruStain FcX (Rat monoclonal) | BioLegend | cat.# 101319 | 1:100 |
| Antibody | Anti-mouse Gr-1 Biotin (Rat monoclonal) | BioLegend | cat.# 108403 | 1:100 |
| Antibody | Anti-mouse/human CD45R/ B220 (Rat monoclonal) | BioLegend | cat.# 103204 | 1:100 |
| Antibody | Anti-mouse CD3 (Rat monoclonal) | BioLegend | cat.# 100243 | 1:100 |
| Antibody | Anti-mouse Ly-6C (Rat monoclonal) | BioLegend | cat.# 128003 | 1:100 |
| Chemical compound, drug | N-Acetyl-L-cysteine (NAC) | Sigma-Aldrich | cat.# A7250 | 2 g/L in drinking water |
| Chemical compound, drug | Iron-Dextran Solution | Sigma-Aldrich | cat.# D8517 | 8 mg |
| Chemical compound, drug | Hemin | Sigma-Aldrich | cat.# 51280 | 20 µM |
| Chemical compound, drug | Ferric Ammonium Citrate (FAC) | Sigma-Aldrich | cat.# F5879 | 150, 50 and 10 µM |
| chemical compound, drug | CORM-A1 | Sigma-Aldrich | cat.# SML0315 | 50 µM |
| Chemical compound, drug | Deferoxamine (DFO) | Sigma-Aldrich | cat.# D9533 | 100 µM |
| Chemical compound, drug | Zinc (II) Protoporphyrin IX (ZnPP) | Sigma-Aldrich | cat.# 691550 | 5, 1 and 0.5 µM |
| Chemical compound, drug | Tunicamycin (Tm) | Sigma-Aldrich | cat.# T7765 | 2.5 µM |
| Chemical compound, drug | Liproxstatin-1 (Lip-1) | Sigma-Aldrich | cat.# SML1414 | 2 µM |
| Chemical compound, drug | EZ-Link Sulfo-NHS Biotin | Thermo Fisher Scientific | cat.# 21217 | 1 mg / 100 µL |
| Commercial assay, kit | LIVE/DEAD Fixable Aqua | Invitrogen | cat.# L34966 | |
| Commercial assay, kit | LIVE/DEAD Fixable Violet | Invitrogen | cat.# L34964 | |
| Commercial assay, kit | CellROX Deep Red Reagent | Invitrogen | cat.# C10422 | |
| Commercial assay, kit | Lipid Peroxidation Assay Kit | Abcam | cat.# ab243377 | |
| Commercial assay, kit | MitoSOX Red | Invitrogen | cat.# M36008 | |
| Commercial assay, kit | Lysosomal Intracellular Activity Assay Kit | Biovision | cat.# K448 | |

*Continued on next page*

*Continued*

| Reagent type (species) or resource | Designation | Source or reference | Identifiers | Additional information |
|---|---|---|---|---|
| Commercial assay, kit | Me4BodipyFL-Ahx3Leu3VS (Fluorescent Proteasome Activity) | R&D Systems | cat.# I-190 | |
| Commercial assay, kit | tetramethylrhodamine ethyl ester - TMRE (Fluorescent Probe) | Sigma-Aldrich | cat.# 87917 | |
| Commercial assay, kit | MitoTracker Green | Invitrogen | cat.# M7514 | |
| Commercial assay, kit | Cal-520 AM Probe | Abcam | cat.# ab171868 | |
| Commercial assay, kit | FerroOrange | DojinD | cat.# F374 | |
| Commercial assay, kit | PROTEOSTAT Aggresome Detection Kit | Enzo Life Sciences | cat.# ENZ-51035–0025 | |
| Commercial assay, kit | ATP Fluorometric Assay Kit | Sigma-Aldrich | cat.# MAK190 | |
| Commercial assay, kit | Iron Assay Kit | Sigma-Aldrich | cat.# MAK025 | |
| Commercial assay, kit | PKH-67 | Sigma-Aldrich | cat.# MIDI67-1KT | |
| Commercial assay, kit | Heme Assay Kit | Sigma-Aldrich | cat.# MAK316 | |
| Commercial assay, kit | SFBC | Biolabo | cat.# 80008 | |
| Commercial assay, kit | UIBC | Biolabo | cat.# 97408 | |
| Commercial assay, kit | Mouse Erythropoietin/EPO Quantikine ELISA Kit | R&D Systems | cat.# MEP00B | |
| Commercial assay, kit | IL-6 Quantikine ELISA Kit | R&D Systems | cat.# M600B | |
| Commercial assay, kit | Mouse Hemopexin ELISA Kit | Abcam | cat.# ab157716 | |
| Commercial assay, kit | Direct-zol RNA Microprep Kit | Zymo Research | cat.# R2062 | |
| Commercial assay, kit | Alexa Fluor 488 Conjugation Kit (Fast) - Lightning-Link | Abcam | cat.# ab236553 | |
| Commercial assay, kit | Alexa Fluor 647 Conjugation Kit (Fast) - Lightning-Link | Abcam | cat.# ab269823 | |
| Commercial assay, kit | RevertAid H Minus Reverse Transcriptase | Thermo Fisher Scientific | cat.# EP0452 | |
| Commercial assay, kit | SG qPCR Master Mix | EURx | cat.# E0401 | |
| Commercial assay, kit | Iron Stain Kit | Sigma-Aldrich | cat.# HT20-1KT | |
| Commercial assay, kit | Iron Stain Kit | Abcam | cat.# ab150647 | |
| Commercial assay, kit | In Situ Cell Death Detextion Kit | Roche | cat.# 11684795910 | |

*Continued on next page*

*Continued*

| Reagent type (species) or resource | Designation | Source or reference | Identifiers | Additional information |
|---|---|---|---|---|
| Other | Standard iron content diet for mice | SAFE | cat.# U8958v0177 | Diet for young and aged mice |
| Other | Reduced iron content diet for mice | SAFE | cat.# U8958v0294 | Diet for aged IR mice |
| Other | Collagenase D | Roche | cat.# 11088882001 | Digestion of spleen for sorting |
| Other | Trizol-LS | Invitrogen | cat.# 10296010 | RNA isolation reagent for RNA-seq |
| Other | Normal Rat Serum | Thermo Fisher Scientific | cat.# 10,710 C | Blocking reagent for flow cytometry |
| Other | MojoSort Streptavidin Nanobeads | BioLegend | cat.# 480016 | Beads for magnetic sorting of RPMs and aggreagtes |
| Other | MojoSort Streptavidin Nanobeads | BioLegend | cat.# 480072 | Beads for magnetic sorting of RPMs and aggreagtes |
| Other | EasyEights EasySep Magnet | STEMCELL | cat.# 18103 | Magnet used for magnetic sorting of RPMs and aggregates |
| Other | LS Separation Column | Milyenyi Biotec | cat.# 130-042-401 | Magnet used for magnetic sorting of aggregates |
| Other | CPDA-1 | Sigma-Aldrich | cat.# C4431 | Anticoagulant used for blood collection for transfusions |
| Other | Accutase | BioLegend | cat.# 423201 | Reagent used for cells detachment *in vitro* |
| Other | Zymosan A BioParticles Alexa Fluor 488 | Invitrogen | cat.# Z23373 | Cargo for *ex vivo* phgocytosis |
| Other | TRIzol Reagent | Invitrogen | cat.# 15596018 | RNA isolation reagent |
| Peptide, recombinant protein | IL-33 | BioLegend | cat.# 580506 | 10 ng/mL |
| Peptide, recombinant protein | Mini-hepcidin (PR73) | Gift from Elizabeta Nemeth, UCLA, USA | https://doi.org/10.1128/IAI.00253-18 | 2 µg/mL |
| Peptide, recombinant protein | Macrophage Colony-Stimulating Factor (M-CSF) | BioLegend | cat.# 576406 | 20 ng/mL |
| Sequence-based reagent | HAMP Forward | This paper | Real-time PCR primers | 5' ATACCAATGCAGAAGAGAAGG-3' |
| Sequence-based reagent | HAMP Reverse | This paper | Real-time PCR primers | 5'-AACAGATACCACACTGGGAA-3' |
| Software, algorithm | FlowJo | FlowJo | v10.8.1 | |
| Software, algorithm | CytExpert | Beckman Coulter | v2.4 | |
| Software, algorithm | GraphPad Prism | GraphPad Software | v9 | |

## Mice

Female C57BL/6 J mice were used for all the experiments and were maintained in specific pathogen-free (SPF) conditions at the Experimental Medicine Centre (Bialystok, Poland). Starting with the age of 4 weeks mice were fed a diet with a standard iron content (200 mg/kg, SAFE #U8958v0177; for Young and Aged), as described before (*Kautz et al., 2008*; *Pagani et al., 2011*) or reduced iron content [25 mg/kg; SAFE #U8958v0294; for Aged iron reduced (IR)]. The diets were from SAFE (Augy, France). Mice were analyzed at 8–10 weeks (Young) and 10–11 months (Aged and Aged IR) of age. For

supplementation with N-Acetyl-L-cysteine (NAC), aging mice (Aged NAC) were supplied with NAC dissolved in drinking water (2 g/L) from 8 weeks of age until 10–11 months of age. Mice were delivered to the local facility at the IIMCB and sacrificed after short acclimatization, or subjected to additional procedures, if applicable. C57BL/6 J for primary cell cultures were maintained in the SPF facility of Mossakowski Medical Research Institute (Warsaw, Poland). All animal experiments and procedures were approved by the local ethical committees in Olsztyn and Warsaw (decisions: WAW2/015/2019; WAW2/149/2019; WAW2/026/2020; WAW2/149/2020).

### Iron dextran injections

Mice were injected into the peritoneum (IP injection) with 8 mg of iron dextran (Sigma-Aldrich, D8517) or an equivalent amount of dextran (Sigma-Aldrich, BCBZ5113). Animals were sacrificed for flow cytometric analysis of RPMs or isolation of cell-free iron-rich splenic aggregates (see details below) at the indicated time points. The procedure was approved by the local ethical committee in Warsaw (decision: WAW2/122/2019).

### Preparation of single-cell suspension from mouse organs

**Bone marrow** cells were harvested by flushing the femur and tibia using a 25 G needle and sterile HBSS medium (Gibco, 14025092). Cells were centrifuged at 600 g for 10 min at 4°C. The **spleen** was excised and mashed through a 70 µm strainer (pluriSelect, 43-50070-51). For FACS and magnetic sorting, the spleen was additionally digested in HBSS medium containing 1 mg/ml Collagenase D (Roche, 11088882001) and 50 U/ml DNase I for 30 min at 37°C. After that cells were washed with cold HBSS and centrifuged at 600 g for 10 min at 4°C. The **liver** was extracted and perfused using Liver Perfusion Medium (Gibco, 17701038). Next, the organ was minced and digested in Liver Digest Medium (Gibco, 17703034) for 30 min at 37°C. After that liver was pressed through a 70 µm strainer in the presence of HBSS. Cells were centrifuged at 50 g for 3 min and the pellet was discarded (hepatocytes). The supernatant was centrifuged at 700 g for 15 min at 4°C. Pellet was resuspended in 5 mL of PBS containing 0,5% BSA and 5 mL of 50% Percoll (Cytiva, 17-0891-01) diluted in PBS. The suspension was centrifuged at 700 g for 30 min at 20°C. **Blood** was collected to a heparin-coated tube *via* heart puncture. Cells were washed with HBSS medium and centrifuged at 400 g for 10 min at 4°C. For **peritoneal cell** isolation, peritoneal cavities were washed with HBSS medium, and the peritoneal fluid was aspirated and filtered through 70 µm strainer, and cells were centrifuged at 600 g for 5 min at 4 °C.

RBCs present in a single-cell suspension were lysed using 1 X RBC Lysis buffer (BioLegend, 420302) for 3 min at 4°C. This step was omitted for analyses of erythroid progenitor cells and RBCs from the spleen or the bone marrow. Next, cells were washed with HBSS and centrifuged at 600 g for 10 min at 4°C. Pellet was prepared for further functional assays and labeling with antibodies.

### Generation of iRPMs and treatments

Sterile harvested mononuclear cells obtained from mouse femurs and tibias were cultured at 37°C in 5% $CO_2$ at $0.5X10^6/1$ mL concentration in RPMI-1640 (Sigma-Aldrich, R2405) supplemented with 10% FBS (Cytiva, SV30160.03), 1 X Penicillin-Streptomycin (Gibco, 15140122), and 20 ng/mL macrophage colony-stimulating factor (M-CSF, BioLegend, 576406). On the 4th and 6th day medium was changed to fresh, supplemented with 20 µM of hemin (Sigma-Aldrich, 51280) and 10 ng/mL of IL-33 (BioLegend, 580506). Assays were performed on the 8th day.

For treatments, ferric ammonium citrate (FAC, Sigma-Aldrich, F5879), CORM-A1 (Sigma-Aldrich, SML0315), deferoxamine (DFO, Sigma-Aldrich, D9533), and mini-hepcidin (PR73, a kind gift from Elizabeta Nemeth, UCLA, USA) (*Stefanova et al., 2018*) were diluted in sterile ddH2O. Zinc (II) Protoporphyrin IX (ZnPP, Sigma-Aldrich, 691550), tunicamycin (Tm, Sigma-Aldrich, T7765) and Liproxstatin-1 (Lip-1, Sigma-Aldrich, SML1414) were diluted in anhydrous DMSO. Hemin (Sigma-Aldrich, 51280) solution was prepared with 0.15 M NaCl containing 10% NH4OH. All reagents after dilution were filtered through a 0.22 µm filter and stored at –20°C except ferric ammonium citrate, which was always freshly prepared. The concentration of compounds and duration of treatments are indicated in the descriptions of the figures.

## Flow cytometric analysis and cell sorting

Cell suspensions of spleens and livers and iRPMs (~1 × 10^7) were stained with LIVE/DEAD Fixable Aqua/Violet (Invitrogen, L34966/L34964) as per the manufacturer's instructions to identify dead cells. After extensive washing, the cells were incubated with Fc block in a dilution of 1:100 in FACS buffer for 10 min at 4 °C. Cells were then stained with fluorophore-conjugated antibodies, dilution of 1:100 to 1:400, depending on the titration, in FACS buffer for 30 min at 4 °C. Cells were washed thoroughly with FACS buffer and subjected to flow cytometry analysis. For analysis of the *splenic RPM population*, the following surface antibodies were used: CD45 (BioLegend, 30-F11), CD45R/B220 (BioLegend, RA3-6B2), F4/80 (BioLegend, BM8), CD11b (BioLegend, M1/70) and TREML4 (BioLegend, 16E5). TER-119 (BioLegend) and CD71 (BioLegend, RI7217) were included for *erythroid cell* analysis. For analysis of *liver Kupffer cell and iRPM populations*, the following surface antibodies were used: CD45 (BioLegend, 30-F11), F4/80 (BioLegend, BM8) and CD11b (BioLegend, M1/70). *Detection of FPN* was performed with a non-commercial antibody that recognizes the extracellular loop of mouse FPN [rat monoclonal, Amgen, clone 1C7 (*Sangkhae et al., 2019*); directly conjugated with Alexa Fluor 488 Labeling Kit (Abcam, ab236553)]. For staining of cell membrane receptors, the following antibodies were used: AXL (Invitrogen), MERTK (BioLegend), Tim4 (BioLegend) and Stab2 (G-Biosciences). For analysis of *circulating and splenic RBCs*, the following surface antibodies were used: CD45 (BioLegend, 30-F11), TER-119 (BioLegend) and CD71 (BioLegend, RI7217). For analysis of *splenic granulocytes*, the following surface antibodies were used: CD45 (BioLegend, 30-F11), CD45R/B220 (BioLegend, RA3-6B2), F4/80 (BioLegend, BM8), CD11b (BioLegend, M1/70) and Gr-1 (BioLegend, RB6-8C5). For analysis of *peritoneal macrophages*, the following surface antibodies were used: CD45 (BioLegend, 30-F11), F4/80 (BioLegend, BM8), MHCII (BioLegend, M5/114.15.2), and CD11b (BioLegend, M1/70). Events were either acquired on Aria II (BD Biosciences) or CytoFLEX (Beckman Coulter) and were analyzed with FlowJo or CytExpert, respectively. For RNA sequencing (RNA-seq) and ATP levels quantification, RPMs were sorted into Trizol-LS (Invitrogen, 10296010) or assay buffer, respectively, using an Aria II cell sorter (BD Biosciences) with an 85 µm nozzle.

## Functional assays and intracellular staining of ferrous iron, proteins and protein aggregates

**Intracellular ROS** (APC channel) levels were determined by using CellROX Deep Red Reagent (Invitrogen, C10422) fluorescence according to the manufacturer's instructions. **Lipid peroxidation** (FITC vs PE channels) was determined with the Lipid Peroxidation Assay Kit (Abcam, ab243377) according to the manufacturer's instructions. **Mitochondria-associated ROS** (PE channel) levels were measured with MitoSOX Red (Invitrogen, M36008) at 2.5 µM for 30 min at 37 °C. **Lysosomal activity** (FITC channel) was determined by using Lysosomal Intracellular Activity Assay Kit (Biovision, K448) according to the manufacturer's instructions. **Proteasomal activity** (FITC channel) was determined using Me4BodipyFL-Ahx3Leu3VS fluorescent proteasome activity probe (R&D Systems, I-190) at 2 µM for 1 hr at 37 °C. **Mitochondria activity** (membrane potential, PE channel) was measured using a tetramethylrhodamine ethyl ester (TMRE) fluorescent probe (Sigma-Aldrich, 87917) at 400 nM for 30 min at 37 °C. **Mitochondrial mass** (FITC/PE channel) was measured by fluorescence levels upon staining with MitoTracker Green (Invitrogen, M7514) at 100 nM for 30 min at 37 °C. **Calcium levels** (FITC channel) were determined by using Cal-520 AM probe (Abcam, ab171868) at 5 µM for 1 hour at 37 °C.

The content of **intracellular ferrous iron** (PE channel) ($Fe^{2+}$) was measured using FerroOrange (DojinD, F374) via flow cytometric analysis. Briefly, surface-stained cells were incubated with 1 µM FerroOrange in HBSS for 30 min at 37 °C, and analyzed directly via flow cytometry without further washing. For **intracellular antibody staining**, surface-stained cells were first fixed with 4% PFA and permeabilized with 0.5% Triton-X in PBS. The cells were then stained with the following primary antibodies for 1 hr at 4 °C: Ferritin Heavy Chain (FTH1, Cell Signaling Technology, 3998), Ferritin Light Chain (FTL, Abcam, ab69090), HO-1 polyclonal antibody (Enzo Life Sciences, ADI-OSA-150) and Ki-67 (BioLegend, 16A8). This was followed by 30 min staining with Alexa Fluor 488 or Alexa Fluor 647 conjugated anti-Rabbit IgG (1:1000 Thermo Fisher Scientific, A-21206). **Protein aggregates (aggresomes)** (PE channel) were stained and measured with PROTEOSTAT Aggresome detection kit (Enzo Life Sciences, ENZ-51035–0025) at concentration 1:2000 for 30 minutes at 37 °C.

The geometric mean fluorescence intensities (MFI) corresponding to the probes/target protein levels were determined by flow cytometry acquired on Aria II (BD Biosciences) or CytoFLEX (Beckman Coulter) and were analyzed with FlowJo or CytExpert, respectively. For the probes that have emissions in PE, TREML4 was excluded, and RPMs were gated as F4/80-high CD11b-dim. For quantifications, MFI of the adequate fluorescence minus one (FMO) controls were subtracted from samples MFI, and data were further normalized.

### ATP levels

Cellular ATP levels in FACS-sorted RPMs (10000 cells/sample) were determined using ATP Fluorometric Assay Kit (Sigma-Aldrich, MAK190), as per the manufacturer's instructions.

### Magnetic sorting of RPMs and isolation of extracellular iron-containing aggregates

$60x10^6$ of mouse splenocytes were incubated for 15 min at 4 ° C in PBS containing 5% Normal Rat Serum (Thermo Fisher Scientific, 10,710 C) and anti-CD16/32 (BioLegend, 101320) antibody in 5 mL round bottom tube. Afterward, cells were labeled with anti-F4/80 (APC, BioLegend, 123116), anti-Ly-6G/Ly-6C (Gr-1) (Biotin, BioLegend, 108403), anti-CD3 (Biotin, BioLegend, 100243), anti-mouse Ly-6C (Biotin, BioLegend, 128003) and anti-B220 (Biotin, BioLegend, 103204) for 20 min in 4 ° C in dark. Next, cells were washed with cold PBS containing 2 mM EDTA, 0.5% BSA (hereafter referred to as 'sorting buffer'), and centrifuged at 600 g for 10 min. Pellet was resuspended in 400 µL of sorting buffer containing 50 µL of MojoSort Streptavidin Nanobeads (BioLegend, 480016) and kept in cold and dark for 15 min. After incubation, an additional 400 µL was added to cells, and the tube was placed on EasyEights EasySep Magnet (STEMCELL, 18103) for 7 min. After that supernatant was transferred to a fresh 5 mL tube and centrifuged at 600 g for 10 min. Pellet was resuspended in 100 µL of sorting buffer and 10 µL of MojoSort Mouse anti-APC Nanobeads (BioLegend, 480072) was added. The suspension was gently pipetted and incubated for 15 min at 4 ° C in the dark. Afterward, the tube was placed on a magnet for 7 min. Next, the supernatant was saved for analysis, and beads with attached F4/80 + cells were washed with sorting buffer and counted under a light microscope with a Neubauer chamber. Cells were pelleted and frozen in liquid nitrogen for further analysis.

For isolation of extracellular iron-containing aggregates, splenocytes were resuspended in HBSS and then carefully layered over Lymphosep (3:1) in a FACS tube, creating a sharp spleen cell suspension-Lymphosep interphase. Leukocytes were sorted out from the supernatant after density centrifugation at 400 g at 20 ° C for 25 min. The pellet comprising mostly RBCs, granulocytes, and extracellular iron-containing aggregates was then washed again with HBSS to remove Lymphosep. The cell pellet was re-suspended in a sorting buffer and then passed through a magnetized LS separation column (Miltenyi Biotec). The iron-containing superparamagnetic cells/aggregates were eluted from the demagnetized column, washed, and re-suspended in a sorting buffer. To achieve a pure yield of extracellular iron-containing aggregates and remove any trace contaminations from superparamagnetic RPMs or other leukocytes, cells expressing F4/80, B220, Gr-1, CD3, and Ly-6C were sorted out using MojoSort magnetic cell separation system as previously described. The remaining material comprising mostly aggregates was washed thoroughly and either pelleted and frozen in liquid nitrogen for further analysis (mass spectrometry and iron/heme measurements) or stained with fluorophore-conjugated antibodies for purity verification with flow cytometry.

### Proteomic analyses of splenic aggregates using label-free quantification (LFQ)

#### Sample preparation

Magnetically-isolated aggregates were dissolved in neat trifluoroacetic acid. Protein solutions were neutralized with 10 volumes of 2 M Tris base, supplemented with TCEP (8 mM) and chloroacetamide (32 mM), heated to 95 °C for 5 min, diluted with water 1:5, and subjected to overnight enzymatic digestion (0.5 µg, Sequencing Grade Modified Trypsin, Promega) at 37 °C. Tryptic peptides were then incubated with Chelex 100 resin (25 mg) for 1 hr at RT, desalted with the use of AttractSPE Disks Bio C18 (Affinisep), and concentrated using a SpeedVac concentrator. Prior to LC-MS measurement, the samples were resuspended in 0.1% TFA, 2% acetonitrile in water.

## LC-MS/MS analysis

Chromatographic separation was performed on an Easy-Spray Acclaim PepMap column 50 cm long ×75 µm inner diameter (Thermo Fisher Scientific) at 45 °C by applying a 90 min acetonitrile gradients in 0.1% aqueous formic acid at a flow rate of 300 nl/min. An UltiMate 3000 nano-LC system was coupled to a Q Exactive HF-X mass spectrometer via an easy-spray source (all Thermo Fisher Scientific). The Q Exactive HF-X was operated in data-dependent mode with survey scans acquired at a resolution of 120,000 at m/z 200. Up to 12 of the most abundant isotope patterns with charges 2–5 from the survey scan were selected with an isolation window of 1.3 m/z and fragmented by higher-energy collision dissociation (HCD) with normalized collision energies of 27, while the dynamic exclusion was set to 30 s. The maximum ion injection times for the survey scan and the MS/MS scans (acquired with a resolution of 15,000 at m/z 200) were 45 and 96ms, respectively. The ion target value for MS was set to 3e6 and for MS/MS to 1e5, and the minimum AGC target was set to 1e3.

## Data processing

The data were processed with MaxQuant v. 1.6.17.0 or v. 2.0.3.0 (*Cox and Mann, 2008*), and the peptides were identified from the MS/MS spectra searched against the reference mouse proteome UP000000589 (https://www.uniprot.org/) using the build-in Andromeda search engine. Raw files corresponding to three replicate samples obtained from Ag isolates and three replicate samples obtained from Y isolates were processed together. Raw files corresponding to two replicate samples corresponding to isolates obtained from control animals (dextran-injected, Dex) and two replicate samples corresponding to isolates obtained from animals subjected to intraperitoneal iron dextran injection (Fe-Dex) were processed together. Cysteine carbamidomethylation was set as a fixed modification and methionine oxidation, glutamine/asparagine deamidation, and protein N-terminal acetylation were set as variable modifications. For in silico digests of the reference proteome, cleavages of arginine or lysine followed by any amino acid were allowed (trypsin/P), and up to two missed cleavages were allowed. LFQ min. ratio count was set to 1. The FDR was set to 0.01 for peptides, proteins and sites. Match between runs was enabled. Other parameters were used as pre-set in the software. Unique and razor peptides were used for quantification enabling protein grouping (razor peptides are the peptides uniquely assigned to protein groups and not to individual proteins). Data were further analyzed using Perseus version 1.6.10.0 (*Tyanova et al., 2016*) and Microsoft Office Excel 2016.

## Data processing and bioinformatics

Intensity values for protein groups were loaded into Perseus v. 1.6.10.0. Standard filtering steps were applied to clean up the dataset: reverse (matched to decoy database), only identified by site, and potential contaminants (from a list of commonly occurring contaminants included in MaxQuant) protein groups were removed. Reporter intensity values were normalized to the tissue weight the aggregates were isolated from and then Log2 transformed. Protein groups with valid values in less than 2 Ag (Ag vs Y dataset) or less than 2 Fe-Dex (Fe-Dex vs Dex dataset) samples were removed. For protein groups with less than 2 valid values in Y or Dex, missing values were imputed from a normal distribution (random numbers from the following range were used: downshift = 1.8StdDev, width = 0.4StdDev). For the Ag vs Y dataset, which contained three biological replicate samples per condition, one-sided Student T-testing (permutation-based FDR = 0.001, S0=1) was performed to return 3290 protein groups with levels statistically significantly greater in Ag samples compared to Y samples. For the Fe-Dex vs Dex dataset, which contained two biological replicate samples per condition, a significance cut-off of Log2 fold change Fe-Dex vs Dex >1.5 was applied to return 2577 protein groups with levels significantly greater in Fe-Dex samples compared to Dex samples. For direct comparisons of these two datasets an analogous significance cut-off (Log2 fold change Ag vs Y>1.5) was applied also to the Ag vs Y dataset to return 3614 protein groups with levels significantly greater in Ag samples compared to Y samples. Annotation enrichment analysis was performed using DAVID (https://david.ncifcrf.gov/) and ShinyGO (http://bioinformatics.sdstate.edu/go/), using FDR = 0,05 as a threshold.

## Proteomic analyses of splenic aggregates using Tandem Mass Tag (TMT) labeling

### Sample preparation

Magnetically isolated aggregates were dissolved in neat trifluoroacetic acid. Protein solutions were neutralized with 10 volumes of 2 M Tris base, supplemented with TCEP (8 mM) and chloroacetamide (32 mM), heated to 95° for 5 min, diluted with water 1:5, and subjected to overnight enzymatic digestion (0.5 µg, Sequencing Grade Modified Trypsin, Promega) at 37 °C. Tryptic peptides were then incubated with Chelex 100 resin (25 mg) for 1 hr at RT, desalted with the use of AttractSPE Disks Bio C18 (Affinisep), TMT-labeled on the solid support (*Myers et al., 2019*), compiled into a single TMT sample and concentrated using a SpeedVac concentrator. Prior to LC-MS measurement, the samples were resuspended in 0.1% TFA, 2% acetonitrile in water.

### LC-MS/MS analysis

Chromatographic separation was performed on an Easy-Spray Acclaim PepMap column 50 cm long ×75 µm inner diameter (Thermo Fisher Scientific) at 45 °C by applying a 120 min acetonitrile gradients in 0.1% aqueous formic acid at a flow rate of 300 nl/min. An UltiMate 3000 nano-LC system was coupled to a Q Exactive HF-X mass spectrometer via an easy-spray source (all Thermo Fisher Scientific). Three samples injections were performed. The Q Exactive HF-X was operated in data-dependent mode with survey scans acquired at a resolution of 60,000 at m/z 200. Up to 15 of the most abundant isotope patterns with charges 2–5 from the survey scan were selected with an isolation window of 0.7 m/z and fragmented by higher-energy collision dissociation (HCD) with normalized collision energies of 32, while the dynamic exclusion was set to 35 s. The maximum ion injection times for the survey scan and the MS/MS scans (acquired with a resolution of 45,000 at m/z 200) were 50 and 96ms, respectively. The ion target value for MS was set to 3e6 and for MS/MS to 1e5, and the minimum AGC target was set to 1e3.

### Data processing

The data were processed with MaxQuant v. 1.6.17.0 (*Cox and Mann, 2008*), and the peptides were identified from the MS/MS spectra searched against the reference mouse proteome UP000000589 (https://www.uniprot.org/) using the build-in Andromeda search engine. Raw files corresponding to 3 replicate injections of the combined TMT sample were processed together as a single experiment/ single fraction. Cysteine carbamidomethylation was set as a fixed modification and methionine oxidation, glutamine/asparagine deamidation, and protein N-terminal acetylation were set as variable modifications. For in silico digests of the reference proteome, cleavages of arginine or lysine followed by any amino acid were allowed (trypsin/P), and up to two missed cleavages were allowed. Reporter ion MS2 quantification was performed with the min. reporter PIF was set to 0.75. The FDR was set to 0.01 for peptides, proteins and sites. Match between runs was enabled and second peptides function was disabled. Other parameters were used as pre-set in the software. Unique and razor peptides were used for quantification enabling protein grouping (razor peptides are the peptides uniquely assigned to protein groups and not to individual proteins). Data were further analyzed using Perseus version 1.6.10.0 (*Tyanova et al., 2016*) and Microsoft Office Excel 2016.

### Data processing and bioinformatics

Reporter intensity corrected values for protein groups were loaded into Perseus v. 1.6.10.0. Standard filtering steps were applied to clean up the dataset: reverse (matched to decoy database), only identified by site, and potential contaminants (from a list of commonly occurring contaminants included in MaxQuant) protein groups were removed. Reporter intensity values were Log2 transformed and normalized by median subtraction within TMT channels. 942 Protein groups with the complete set of valid values were kept. Student T-testing (permutation-based FDR = 0.05, S0=0.1) was performed on the dataset to return 70 protein groups, which levels were statistically significantly changed in Ag vs IR samples. Annotation enrichment analysis was performed using DAVID (https://david.ncifcrf.gov/) and ShinyGO (http://bioinformatics.sdstate.edu/go/), using FDR = 0,05 as a threshold.

## Identification of proteins in RPMs

### Sample preparation

Magnetically isolated RPMs were isolated from spleens of two female 8-weeks-old C57BL/6 J. Cells were lysed in RIPA buffer. Proteins were precipitated with chloroform/methanol, protein pellet washed with methanol, and then reconstituted in 100 mM HEPES pH 8.0 containing 10 mM TCEP and 10 mM chloroacetamide. Proteins were subjected to overnight enzymatic digestion (Sequencing Grade Modified Trypsin, Promega) at 37 °C. Tryptic peptides were acidified with trifluoroacetic acid (final conc. 1%), desalted with the use of AttractSPE Disks Bio C18 (Affinisep), and concentrated using a SpeedVac concentrator. Prior to LC-MS measurement, the samples were resuspended in 0.1% TFA, 2% acetonitrile in water.

### LC-MS/MS analysis

Chromatographic separation was performed on an Easy-Spray Acclaim PepMap column 50 cm long ×75 µm inner diameter (Thermo Fisher Scientific) at 55 °C by applying a 90 min (protein aggregates) or a 180 min (RPM extract) acetonitrile gradients in 0.1% aqueous formic acid at a flow rate of 300 nl/min. An UltiMate 3000 nano-LC system was coupled to a Q Exactive HF-X mass spectrometer via an easy-spray source (all Thermo Fisher Scientific). The Q Exactive HF-X was operated in data-dependent mode with survey scans acquired at a resolution of 120,000 at m/z 200. Up to 12 of the most abundant isotope patterns with charges 2–5 from the survey scan were selected with an isolation window of 1.3 m/z and fragmented by higher energy collision dissociation (HCD) with normalized collision energies of 27, while the dynamic exclusion was set to 30 s. The maximum ion injection times for the survey scan and the MS/MS scans (acquired with a resolution of 15,000 at m/z 200) were 45 and 96ms, respectively. The ion target value for MS was set to 3e6 and for MS/MS to 1e5, and the minimum AGC target was set to 1e3.

### Data processing

The data were processed with MaxQuant 2.1.3.0, and the peptides were identified from the MS/MS spectra searched against Uniprot Mouse Reference Proteome (UP000000589) using the built-in Andromeda search engine. Cysteine carbamidomethylation was set as a fixed modification and methionine oxidation, glutamine/asparagine deamination, as well as protein N-terminal acetylation were set as variable modifications. For in silico digests of the reference proteome, cleavages of arginine or lysine followed by any amino acid were allowed (trypsin/P), and up to two missed cleavages were allowed. The FDR was set to 0.01 for peptides, proteins and sites. Match between runs was enabled. Other parameters were used as pre-set in the software. Proteins were identified and intrasample quantified using iBAQ algorithm available in MaxQuant.

## Measurement of cellular iron levels

Determination of cellular total iron levels in magnetically-sorted RPMs and splenic protein aggregates was carried out using the Iron Assay Kit (Sigma-Aldrich, MAK025) according to the manufacturer's instructions, and as shown previously (*Folgueras et al., 2018*). For the measurement of total iron in RPMs and splenic aggregates volumes of buffers and reagents were decreased proportionally. Absorbance at 593 nm was measured with Nanodrop ND-1000 Spectrophotometer (Thermo Fisher Scientific). For RPMs, iron concentrations (ng/µL) were calculated from the standard curve and normalized to the number of cells in each sample. For the aggregates, iron amounts (µg/g fresh tissue) were calculated from the standard curve and normalized to the weight of fresh tissue.

## *In vivo* RBC lifespan

EZ-Link Sulfo-NHS Biotin (Thermo Fisher Scientific, 21217) was dissolved in sterile PBS to a final concentration of 1 mg per 100 µL and filtered through a 0.1 µm filter (Millipore, SLVV033RS). A day before the first blood collection, 100 µL of the sterile solution was injected intravenously into mice. On days 0, 4, 11, 18, and 25 approximately 10 µL of whole blood was collected from the tail vein with heparinized capillary to a tube containing HBSS. RBCs were centrifuged at 400 g for 5 min at 4 °C. Each sample was resuspended in 250 µL of HBSS containing 5% normal rat serum (Thermo Fisher Scientific). Then 2 µL of fluorescently labeled anti-TER-119 and streptavidin was added to the

suspension. Fluorescent streptavidin was omitted for FMO samples in each group. After incubation at 4°C for 30 min, samples were centrifuged and resuspended with HBSS. The percentage of biotinylated erythrocytes was determined by flow cytometry.

## Preparation of stressed erythrocytes for erythrophagocytosis assays

Preparation and staining of stressed RBCs (sRBCs) were performed as described before (*Theurl et al., 2016*), with some modifications. *Preparation of RBCs*: Mice were sacrificed and whole blood was aseptically collected *via* cardiac puncture to CPDA-1 solution (Sigma-Aldrich, C4431). The final concentration of CPDA-1 was 10%. Whole blood obtained from mice was pooled and then centrifuged at 400 g for 15 min at 4°C. Plasma was collected and filtered through a 0.1 μm filter and stored at 4°C. RBCs were resuspended in HBSS and leukoreduced using Lymphosep (Biowest, L0560-500). Cells were washed with HBSS and then heated for 30 min at 48°C while continuously shaking, generating sRBC. *Staining of sRBCs*: $1 \times 10^{10}$ RBC were resuspended in 1 ml diluent C, mixed with 1 ml diluent C containing 4 μM PKH-67 (Sigma-Aldrich, MIDI67-1KT) and incubated in dark for 5 min in 37°C, the reaction was stopped by adding 10 mL HBSS containing 2% FCS and 0.5% BSA. Staining was followed by two washing steps with HBSS. For *in vitro* and *ex vivo* erythrophagocytosis assay cells were resuspended in RPMI-1640 and counted.

For in vivo approach, RBCs were resuspended to 50% hematocrit in previously collected and filtered plasma.

### *In vitro* erythrophagocytosis

Stained and counted sRBCs were added to iRPMs on 12-well plates in 10-fold excess for 1.5 h in 37°C, 5% $CO_2$ on the 8th day after seeding. After that cells were extensively washed with cold PBS to discard not engulfed sRBCs. Next, cells were detached with Accutase (BioLegend, 423201), transferred to a round bottom tube, washed with HBSS, and centrifuged at 600 g for 5 min. Cells were labeled with antibodies and analyzed by flow cytometry.

### *In vivo* erythrophagocytosis

Mice were injected into the tail vein with 100 μL of RBCs resuspended in plasma to 50% hematocrit. Mice were maintained for 1.5 hr in cages with constant access to water and food. After that time animals were sacrificed for organ isolation.

### *Ex vivo* phagocytosis and erythrophagocytosis

$10 \times 10^6$ splenocytes were resuspended in a 5 mL round bottom tube in 200 μL of warm complete RPMI-1640. Fluorescent sRBCs or fluorescent Zymosan A particles (Invitrogen, Z23373) were added to cells at ratio 10:1 (sRBCs/Zymosan: Cells) for 1.5 hr at 37°C, 5% $CO_2$. Afterward, cells were centrifuged at 600 g for 5 min. Excess of sRBCs was lysed using 1 X RBCs lysis buffer, cells were washed with HBSS and centrifuged. Next, cells were labeled with fluorescent antibodies and analyzed by flow cytometry.

## Heme content analysis

For the *extracellular splenic heme content*, the whole spleen was weighted, quickly dissected, and gently mashed through a 100 μm strainer in the presence of 3 mL HBSS. After that suspension was centrifuged at 400 g for 10 min at 4°C. A splenocyte pellet was used for other purposes and the supernatant was transferred to a 1,5 mL tube and centrifuged at 1000 g for 10 min at 4°C to remove the rest of the cells and membranes. *Splenic aggregates* for heme measurements were isolated as previously described and were resuspended directly in Heme Reagent from Heme Assay Kit (Sigma-Aldrich, MAK316). *Blood from the portal vein* was collected to the heparin-coated tube, centrifuged at 400 g for 10 min at 4°C and plasma was transferred to the 1,5 mL tube. Heme concentrations were measured using Heme Assay Kit (Sigma-Aldrich, MAK316) according to manufacturer instructions. Absorbance was measured at 400 nm. The amount of heme was calculated against Heme Calibrator and additionally normalized to the initial weight of fresh spleens (for splenic extracellular heme content and splenic aggregates).

## Transferrin saturation and tissue iron measurements

Serum iron and unsaturated iron-binding capacity were measured with SFBC (Biolabo, 80008) and UIBC (Biolabo, 97408) kits according to manufacturer protocols. Transferrin saturation was calculated

using the formula SFBC/(SFBC +UIBC)x100. For measurement of tissue non-heme iron content, the bathophenanthroline method was applied and calculations were made against tissue dry weight, as described previously (*Torrance and Bothwell, 1968*).

## Erythropoietin (EPO), IL-6 and hemopexin (HPX) measurement with ELISA

The plasma levels of erythropoietin, IL-6 and hemopexin were measured by Mouse Erythropoietin/ EPO Quantikine ELISA Kit (R&D Systems, MEP00B), IL-6 Quantikine ELISA Kit (R&D Systems, M6000B) and Mouse Hemopexin ELISA Kit (Abcam, ab157716) according to the manufacturer's instructions. The optical density was measured on a microplate reader at a wavelength of 450 nm with wavelength correction set to 570 nm.

## RNA isolation

RNA from sorted cells was isolated from TRIzol LS Reagent (Invitrogen, 10296028) using Direct-zol RNA Microprep Kit (Zymo Research, R2062) according to the manufacturer's instructions. RNA from tissues was isolated from TRIzol Reagent (Invitrogen, 15596018) following the guidelines of the manufacturer protocol.

## Reverse transcription and qRT-PCR

cDNA was synthesized with RevertAid H Minus Reverse Transcriptase (Thermo Fisher Scientific, EP0452) according to manufacturer guidelines. Real-time PCR was performed by using SG qPCR Master Mix (EURx, E0401) and HAMP gene primers (Forward 5'-ATACCAATGCAGAAGAGAAGG-3', Reverse 5'-AACAGATACCACACTGGGAA-3') as described in manufacturer protocol. qRT-PCR was run on LightCycler 96 System (Roche).

## Histological and histochemical analysis

Following fixation in 10% formalin for 24 h, spleens were stored in 70% ethanol before further preparation. The tissue was embedded in paraffin and 7 µm cross-sections were cut with a microtome (Reichert-Jung, Germany). The sections were stained with hematoxylin and eosin. Slides were examined by light microscopy (Olympus, type CH2). Non-heme iron staining of spleen samples was analyzed using Iron Stain Kit (Sigma-Aldrich, HT20-1KT). Sections were prepared as described above. After mounting on glass slides, sections were deparaffinized, incubated with a working solution containing Perls' Prussian Blue for 30 min, counterstained with pararosaniline solution for 2 min, and analyzed under standard light microscopy (Olympus CH2).

## Detection of apoptotic cell death

Spleens were dissected, fixed in 4% paraformaldehyde (Sigma-Aldrich) in phosphate-buffered saline (PBS) (Sigma-Aldrich) at 4 °C for 24 hr and then washed two times for 30 min in PBS, soaked in 12.5% sucrose (Bioshop) for 1.5 hr and in 25% sucrose (Bioshop) for not less than 24 hr. All incubations were performed at 4 °C. Next, tissues were washed in PBS, embedded in Tissue-Tek compound, frozen in liquid nitrogen and sectioned into 20 µm slices using a cryostat (Shandon, UK). Apoptotic cell death was detected using In Situ Cell Death Detection Kit (Roche, 11684795910) according to the manual instruction with small modifications. The sections were washed in PBS for 10 min and permeabilized in 0.1% Triton X–100 in 0.1% sodium citrate (Sigma-Aldrich) for 30 min at room temperature. Then the sections were washed 3 times with PBS and incubated with PBS for 30 min. After it, sections were incubated with 50 µl of solution of Enzyme Solution (Blue) and Label Solution (Purple) (mix in a 1: 9 ratio) from the kit at 37 °C for 2 hr. Finally, the sections were washed three times for 5 min in PBS at RT and mounted using Vectashield with 49,6-diamidine-2- phenylindole (DAPI; Vector Labs). Slides were analyzed with a Zeiss LSM 510 Meta confocal microscope (Carl Zeiss, Jena, Germany) using the 60 x objective.

## Transmission electron microscopy (TEM) of the spleen

Fresh samples of the spleen, about 3 square mm, were fixed in 2.5% glutaraldehyde for 24 hr at 4 °C, then washed in PBS and postfixed with 1% osmium tetroxide for 1 hr. After washing with water they were incubated with 1% aqueous uranyl acetate for 12 hr at 4 °C. Next, samples were dehydrated

at room temperature with increasing concentrations of ethanol, infiltrated with epoxy resin (Sigma-Aldrich, 45-359-1EA-F) and subjected for polymerization for 48 hr at 60 ° C. Polymerized resin blocks were trimmed with a tissue processor (Leica EM TP), cut with an ultramicrotome (EM UC7, Leica) for ultrathin sections (65 nm thick), and collected on nickel grids, mesh 200 (Agar Scientific, G2200N). Specimen grids were examined with a transmission electron microscope Tecnai T12 BioTwin (FEI, Hillsboro, OR, USA) equipped with a 16 megapixel TemCam-F416 (R) camera (TVIPS GmbH) at in-house Microscopy and Cytometry Facility.

## Transcriptome analysis by RNA-seq

To prepare libraries from FACS-sorted RPMs (at least 100,000 cells/sample), we used the previously described Smart-seq2 protocol (*Picelli et al., 2013*), suitable for low-input total mRNA sequencing. The quality of RNA and material during the preparation of libraries was checked by Bioanalyzer. The samples were sequenced on NextSeq500 (Illumina) with 75 bp single-end reads, with ~50 million reads/sample. RNAseq was performed at GeneCore at EMBL (Heidelberg, Germany). The quality of the reads was assessed with FastQC software [https://www.bioinformatics.babraham.ac.uk/projects/fastqc/]. Reads were mapped to the *Mus musculus* genome assembly GRCm38(mm10) with HISAT2 software (RRID:SCR_015530; version 2.2.1) [http://daehwankimlab.github.io/hisat2/] on default parameters. Then, the mapped reads were counted into Ensembl annotation intervals using HTSeq-count software [https://www.ncbi.nlm.nih.gov/labs/pmc/articles/PMC4287950/]. Differentially expressed genes were estimated using DESeq2 software [https://www.ncbi.nlm.nih.gov/labs/pmc/articles/PMC4302049/] with default parameters. Genes with p-adjusted <0.05 were regarded as differentially expressed and included in further analysis. Functional analysis was conducted using ClusterProfiler [https://doi.org/10.1016/j.xinn.2021.100141].

## Statistical analysis

Female mice of the same age were randomly attributed to experimental groups (diets or NAC administration). Mouse-derived samples were collected randomly within groups and often assessed/measured randomly, and groups were harvested in a different order in individual experiments. The investigators were not blinded during experiments and assessments. Sample size (typically 4–7 mice/group or independent biological cell-based experiments) was determined based on power analysis, prior experience of performing similar experiments, and previously published papers in the field of RPM biology (*Akilesh et al., 2019*; *Lu et al., 2020*; *Ma et al., 2021*; *Okreglicka et al., 2021*). Statistical analysis was performed with GraphPad Prism (GraphPad software, Version 9). Data are represented as mean ± SEM, unless otherwise specified. ROUT method was applied (in rare cases) to identify and remove outliers. For all experiments, $\alpha=0.05$. When two groups were compared two-tailed unpaired Welch's t-test was applied, whereas for multiple comparisons, the One-Way Analysis of Variance (ANOVA) test was performed. For ANOVA, Dunnett's Multiple Comparison test was used for experiments comparing multiple experimental groups to a single control, while post-hoc Tukey's test was used to compare multiple experimental groups. The number of mice/samples per group or the number of independent cell-based experiments are shown in the figures or indicated in figure legends. Results were considered as significant for $p < 0.05$ (* - $p < 0.05$, ** - $p < 0.01$, *** -$p < 0.001$, ****- $p < 0.0001$).

## Acknowledgements

We thank the GeneCore team (EMBL, Heidelberg) for performing RNA sequencing. We thank Tara Arvedson (Amgen Inc.Inc, USA) for the anti-ferroportin antibody, Elizabeta Nemeth (UCLA, USA) for PR73, and Ewelina Szymańska (IIMCB, Warsaw) for Liproxstatin-1. Many thanks to Agnieszka Popielska and Anna Kosson, and the staff of the Experimental Medicine Centre (Bialystok, Poland) and Mossakowski Medical Research Institute (Warsaw, Poland) for their technical support. We thank Dr. Dorota Stadnik for the preparation and measurement of the proteomic samples and Dawid Hatala for his assistance in histological analyses. We acknowledge the help of Raffaella Gozzellino in the initial phase of project conceptualization as well as the advice from Malgorzata Piechota (IIMCB, Warsaw) on heat shock application in cell culture. We thank Eryk Szymanski for his help in the bioinformatic analysis of the proteomics data and the generation of Venn diagrams. We are grateful to Florent Ginhoux and Zhaoyuan Liu [Singapore Immunology Network (SIgN)] for sharing their raw experimental data and

assisting in their re-analysis. The model figure was prepared using https://biorender.com/. Proteomic measurements were performed at the Proteomics Core Facility, IMol Polish Academy of Sciences utilizing the equipment funded by the 'Regenerative Mechanisms for Health' project MAB/2017/2 within the International Research Agendas program of the Foundation for Polish Science, co-financed by the European Union under the European Regional Development Fund. WP and KMS acknowledge internal IIMCB funding for inter-lab projects. WP acknowledges funding from Norwegian Financial Mechanism 2014–2021 and operated by the Polish National Science Center under the project contract no UMO-2019/34 /H/NZ3/00691. KMS acknowledges funding from the National Science Centre Sonata Bis grant (UMO-2020/38/E/NZ4/00511).

## Additional information

### Funding

| Funder | Grant reference number | Author |
|---|---|---|
| National Science Centre | Sonata Bis grant (UMO-2020/38/E/NZ4/00511) | Raghunandan Mahadeva Komal Chouhan Marta Niklewicz |
| Norwegian Financial Mechanism 2014-2021/ Polish National Science Centre | UMO-2019/34/H/NZ3/00691 | Wojciech Pokrzywa |
| Foundation for Polish Science | International Research Agendas program MAB/2017/2 | Remigiusz Serwa |

The funders had no role in study design, data collection and interpretation, or the decision to submit the work for publication.

### Author contributions

Patryk Slusarczyk, Pratik Kumar Mandal, Conceptualization, Formal analysis, Investigation, Visualization, Methodology, Writing – original draft; Gabriela Zurawska, Matylda Macias, Formal analysis, Investigation, Visualization, Methodology; Marta Niklewicz, Komal Chouhan, Magdalena Cybulska-Lubak, Sylwia Herman, Investigation; Raghunandan Mahadeva, Aneta Jończy, Investigation, Visualization; Aleksandra Szybinska, Formal analysis, Investigation, Methodology; Olga Krawczyk, Formal analysis, Visualization, Methodology; Michal Mikula, Formal analysis, Validation; Remigiusz Serwa, Formal analysis, Funding acquisition, Validation, Visualization, Methodology; Małgorzata Lenartowicz, Formal analysis, Validation, Investigation, Visualization, Methodology; Wojciech Pokrzywa, Conceptualization, Resources, Formal analysis, Supervision, Funding acquisition, Validation, Investigation, Visualization, Methodology, Project administration, Writing – review and editing; Katarzyna Mleczko-Sanecka, Conceptualization, Resources, Formal analysis, Supervision, Funding acquisition, Validation, Investigation, Visualization, Methodology, Writing – original draft, Project administration, Writing – review and editing

### Author ORCIDs

Patryk Slusarczyk (iD) http://orcid.org/0000-0003-2383-3630
Pratik Kumar Mandal (iD) http://orcid.org/0000-0002-1566-5641
Gabriela Zurawska (iD) http://orcid.org/0000-0003-1465-8957
Marta Niklewicz (iD) http://orcid.org/0000-0002-3407-795X
Komal Chouhan (iD) http://orcid.org/0000-0003-4769-8787
Raghunandan Mahadeva (iD) http://orcid.org/0000-0001-5865-6430
Aneta Jończy (iD) http://orcid.org/0000-0002-4333-6853
Magdalena Cybulska-Lubak (iD) http://orcid.org/0000-0002-1029-3156
Sylwia Herman (iD) http://orcid.org/0000-0001-9750-0746
Michal Mikula (iD) http://orcid.org/0000-0003-3447-7328
Remigiusz Serwa (iD) http://orcid.org/0000-0002-4684-3754
Małgorzata Lenartowicz (iD) http://orcid.org/0000-0003-4714-0783

Wojciech Pokrzywa http://orcid.org/0000-0002-5110-4462
Katarzyna Mleczko-Sanecka http://orcid.org/0000-0001-9095-9597

### Ethics

All animal experiments and procedures were approved by the local ethical committees for animal care and use in Olsztyn and Warsaw (II LKE) (decisions: WAW2/015/2019; WAW2/149/2019; WAW2/026/2020; WAW2/149/2020; WAW2/122/2019).

### Decision letter and Author response

Decision letter https://doi.org/10.7554/eLife.79196.sa1
Author response https://doi.org/10.7554/eLife.79196.sa2

## Additional files

### Supplementary files

• MDAR checklist

### Data availability

RNA sequencing data are deposited in the GEO repository (under accession no: GSE199879). Mass spectrometry proteomics data were deposited to the ProteomeXchange Consortium via the PRIDE partner repository with the dataset identifiers: PXD032900 and PXD038660. All other numerical data used to generate the figures are provided as Source data files.

The following datasets were generated:

| Author(s) | Year | Dataset title | Dataset URL | Database and Identifier |
|---|---|---|---|---|
| Slusarczyk P, Mandal PK, Zurawska G, Niklewicz M, Chouhan K, Macias M, Szybinska A, Cybulska M, Krawczyk O, Herman S, Mikula M, Lenartowicz M, Pokrzywa W, Mleczko-Sanecka K | 2022 | Impaired iron recycling from erythrocytes is an early hallmark of aging | https://www.ncbi.nlm.nih.gov/geo/query/acc.cgi?acc=GSE199879 | NCBI Gene Expression Omnibus, GSE199879 |
| Slusarczyk P, Mandal PK, Zurawska G, Niklewicz M, Chouhan K, Macias M, Szybinska A, Cybulska M, Krawczyk O, Herman S, Mikula M, Lenartowicz M, Pokrzywa W, Mleczko-Sanecka K | 2022 | Impaired iron recycling from erythrocytes is an early hallmark of aging | http://proteomecentral.proteomexchange.org/cgi/GetDataset?ID=PXD032900 | ProteomeXchange, PXD032900 |
| Slusarczyk P, Mandal PK, Zurawska G, Niklewicz M, Chouhan K, Mahadeva R, Jonczy A, Macias M, Szybinska A, Cybulska M, Krawczyk O, Herman S, Mikula M, Lenartowicz M, Pokrzywa W, Mleczko-Sanecka K | 2022 | Aging-triggered iron-rich insoluble protein aggregates in the spleen originate chiefly from red pulp macrophages | http://proteomecentral.proteomexchange.org/cgi/GetDataset?ID=PXD038660 | ProteomeXchange, PXD038660 |

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
