## [Editor Report]

Slusarczyk et al. present a well written manuscript focused on understanding the mechanisms underlying aging of erythrophagocytic macrophages in the spleen (RPM) and its relationship to iron loading with age. Importantly, the manuscript demonstrates that RPM erythrophagocytic capacity is diminished with age, restored in iron restricted diet fed aged mice. The main conclusion of the manuscript points to accumulation of unavailable insoluble forms of iron as both causing and resulting from RPM failure, likely a consequence of decreased ferroportin expression on RPMs in the spleen.

---

## [Decision Letter]

**Decision letter after peer review:**

Thank you for submitting your article "Impaired iron recycling from erythrocytes is an early hallmark of aging" for consideration by *eLife*. Your article has been reviewed by 3 peer reviewers, one of whom is a member of our Board of Reviewing Editors, and the evaluation has been overseen by a Reviewing Editor and Carlos Isales as the Senior Editor. The following individual involved in review of your submission has agreed to reveal their identity: Iqbal Hamza (Reviewer #2).

Essential revisions:

Slusarczyk et al. present a comprehensive and extensively documented examination of the effects of aging and cumulative iron-mediated damage on the function of iron-recycling (red pulp) macrophages in the murine spleen. Although the conclusions are not unexpected, the detailed characterization of the functional impairment of this important macrophage population is important and novel. Taken together, we find the manuscript to be of significant potential interest to warrant publication in *eLife*. However, based on the reviewers' comments, several conceptual and technical elements require resolution prior to further consideration and prevent the acceptance of the manuscript in its current form. The main points to be addressed are listed below and a full listing of reviewers' comments can be found in the individual reviews.

1) There is no clear explanation of why iron increases during aging although the authors appear to be saying that iron accumulation is both the cause of and a consequence of decreased RPM erythrophagocytic capacity. By what mechanism does iron restriction benefit RPM aging?

2) Clarification of whether there is shortage of RPMs with aging because of an insufficient recruitment from the circulating pool or there is a functional deficit. What is the underlying mechanism for lower erythrophagocytosis?

3) The proposed model is that during aging both EP and HO-1 expression decreases in RPMs but iron and ferroportin levels are elevated making for a discrepancy with elevated LIP in RPMs under these conditions.

4) Some technical elements, specifically the validation of antibodies for flow and the methods for free heme measures, require further clarification.

*Reviewer #2 (Recommendations for the authors):*

1. Since major conclusions are derived from alterations in various iron/heme metabolism protein levels, it is critical to show that these antibodies (ferritin, HO1, FPN) actually are specific and work in flow cytometry experiments. I am unware of whether these Abs were ever authenticated for flow/FACS analyses.

2. What is the underlying mechanism for lower EP? The authors propose that aging RPMs have decreased lysosomal activity, lower H ferritin, increase LIP and oxidative stress. What is the cause and effect for RPMs from aged mice to behave this way?

3. Figure 2: FPN is high in aged RBCs while Hamp is high in the liver. What is the explanation for this phenomenon? Does BMP6 mRNA increase in the liver too? Or is high Hamp level the result of low-grade inflammation (in aged mice) and consequently the activation of the Stat3 pathway?

4. Does the iron accumulation in aged RPMs due to iron accumulation in aged RBCs?

5. Page 8: Why does Fe restriction decrease ROS production in RBCs? (Figure 2CandD)

6. Page 8: How was heme determined and quantified? How do you know that heme is extracellular? How do you know it is "free"? (Figure 2F)

7. Page 8, line 42,: … our data suggest that reduced EP rate during aging may promote the splenic retention.." I would think that this means that there is higher ROS in red cells from aged mice which could pre-dispose them for preemptive clearance by RPMs, which are unhealthy to begin with in aged mice.

8. Page 9, line 25: "… these aggregates contain large amounts of total iron (Figure 2J) and, interestingly, heme (Figure 2K)". The numbers (fold-change) don't match with total Fe and heme in Figure 1A, even if these were aggregates while the others are total.

9. Page 12, lines 35/36: "… we propose that their formation limits the bioavailability of iron for further systemic utilization".

How does this reconcile with the fact that there is 200ppm dietary Fe to overcome Fe bioavailability? Given that the authors have only looked at the spleen (and in some figures, also the liver), how does the bone marrow, the major erythropoietic organ, deal with this "limited Fe availability"?

If the authors hypothesize that heme-iron deposits are insoluble thus limiting iron bioavailability for systemic utilization, then why is the anemia not corrected in aged IR mice if there is less heme-iron aggregates, and thus, presumably, more available iron?

10. Page 18, line 2: "… causative link between iron loading…".

But the "iron-loaded" cells in vivo have low Ftn levels. Wouldn't this mean that the reason for their perturbation is something else?

11. The justification for use of only female mice is inadequate. We find major differences between male and female mice in iron/heme metabolism and macrophage recycling.

[Editors’ note: further revisions were suggested prior to acceptance, as described below.]

Thank you for resubmitting your work entitled "Impaired iron recycling from erythrocytes is an early hallmark of aging" for further consideration by *eLife*. Your revised article has been evaluated by Carlos Isales (Senior Editor) and a Reviewing Editor.

The manuscript has been improved but there are some remaining issues that need to be addressed, as outlined below:

1) While the authors explained clearly in the response to reviewers regarding point 1 of the essential revision, the explanation does not appear evident or as clear in the revised manuscript. Can the authors note where this is in the abstract, introduction, or discussion? There should be a sentence somewhere in the abstract or discussion or both that says "inflammation with aging leads to increasing hepcidin which suppresses ferroportin, leading to iron accumulation, aggregate formation, and RPM depletion/dysfunction." Furthermore, aggregates correlate with RPM loss in which the only conclusion is that monocyte recruitment does not compensate sufficiently for RPM loss with aging. Please modify in the Discussion on Page 16. The authors are hedging through the manuscript about what is the primary cause and what effect, and it is unclear what causes the initial inflammatory trigger (maybe just accumulated inefficiency of EP?). Removing some of the data and discussion that is afield and focusing on the main thrust of this manuscript would help enormously with readability. 6 figures with 8-16 panels each detract from understanding the main points. Please edit (suggestions made here and below).

2) There appears to be a discrepancy between ferritin decrease with aging and the expectation that it is increased in inflammation, aging, and Figure 7. The authors note that they think ferritin is aggregated but labile iron in RBCs is increased in the spleen from IR-aged mice (Figure 2F). Please edit Figure 7 accordingly, explain the findings in Figure 2F, and add to the Discussion that how ferritin aggregation is affected by excess iron during aging remains to be explored.

3) IR diet does not rescue anemia or alter RNA seq data and the use of agents to remove macrophages altogether does not worsen anemia in mouse models of the disease. Taken together, this raises doubt regarding the central role of RPMs in anemia of aging. While the data is strong overall, there is a possibility that the findings are related more to the standard mouse chow which is a super-physiological iron diet. This must be addressed as a weakness of the study since what the authors actually demonstrate is a comparison between the long-term effects of an iron overload diet and a normal iron diet on RPMs. The manuscript can be modified toward this conceptually or include this idea in the Discussion section since it may not be relevant for normal aging.

---

## [Author Response]

Essential revisions:Slusarczyk et al. present a comprehensive and extensively documented examination of the effects of aging and cumulative iron-mediated damage on the function of iron-recycling (red pulp) macrophages in the murine spleen. Although the conclusions are not unexpected, the detailed characterization of the functional impairment of this important macrophage population is important and novel. Taken together, we find the manuscript to be of significant potential interest to warrant publication in eLife. However, based on the reviewers' comments, several conceptual and technical elements require resolution prior to further consideration and prevent the acceptance of the manuscript in its current form. The main points to be addressed are listed below and a full listing of reviewers' comments can be found in the individual reviews.1) There is no clear explanation of why iron increases during aging although the authors appear to be saying that iron accumulation is both the cause of and a consequence of decreased RPM erythrophagocytic capacity. By what mechanism does iron restriction benefit RPM aging?

We are proposing that intracellular iron accumulation progresses first, chiefly due to ferroportin downregulation, which leads to the depletion of RPMs and the formation of iron-rich extracellular iron-rich protein aggregates. More explanation can be found below (R1 Ad. 1). In addition, we show that iron loading suppresses the erythrophagocytic activity of RPMs, hence further contributing to their impairment during aging.

We believe that iron restricted diet affects RPM aging mainly in two ways. First, it derepresses ferroportin levels and fully prevents iron deposition in RPMs (Figure 1F, H and G). This limits their defective phagocytic activity (Figure 1K; as we also demonstrated in vitro via DFO iron chelation, Figure 6A) and limits their damage (Figure 5C). The importance of the hepcidin-ferroportin axis during RPM aging is discussed in our replies below (please, see R1 Ad. 1; R3 Ad). Furthermore, we noticed that among 55 genes that are differentially regulated between standard and iron-restricted diets, 9 genes (Eps15, Nox1, Ddit4, Cep192, Alms1, Met, Dusp2, Kif5b, Tpra1) are involved in proliferation control (Figure 5—figure supplement 1). Their response uniformly implies an increased proliferative capacity in response to restricted iron feeding. Consistently, we now found that RPMs derived from mice on the IR diet showed higher levels of Ki67, a marker of proliferative cells (Figure 5D). A similar phenomenon was previously shown to drive RPMs recovery after massive erythrophagocytosis-driven depletion (Youssef et al., 2018) and hence can be considered a beneficial mechanism that contributes to RPM niche replenishment.

2) Clarification of whether there is shortage of RPMs with aging because of an insufficient recruitment from the circulating pool or there is a functional deficit.

We addressed this comment in detail in point R1 Ad. 2. We re-analyzed data from Liu et al. (Liu et al., 2019) and we provided the outcome as a new Figure 5E. It clearly shows that the shortage of RPMs during aging is driven by the depletion of RPMs of embryonic origin, not by insufficient recruitment from the monocyte pool. At the age of 10 months, we have not observed any robust change in the population of monocytes or pre-RPMs in the spleen (Figure 5—figure supplement 2A and B), and the numbers of granulocytes showed a mild non-significant decrease (Figure 5—figure supplement 2C).

Furthermore, we now measured in situ apoptosis marker and found a clear sign of apoptosis in the red pulp of aged mice, a phenotype that is less pronounced in mice on an IR diet (Figure 5O). This is consistent with the observation that apoptosis markers can be elevated in tissues upon ferroptosis induction (Friedmann Angeli et al., 2014) and that the proteotoxic stress, which we now emphasized better in the context of RPM aging, may also lead to apoptosis (Brancolini and Iuliano, 2020). Taken together, we strongly believe that the functional defect of embryonically-derived RPMs chiefly contributes to their shortage during aging.

What is the underlying mechanism for lower erythrophagocytosis?

As we now show in additional panels in Figure 6E-G, iron loading reduced protein expression levels of the apoptotic cell receptors MERTK, AXL and TIM4, which are expressed in high levels in RPMs (Slusarczyk and Mleczko-Sanecka, 2021) and thus likely contribute to erythrophagocytosis. Investigation of their exact roles in RBC clearance would need to be addressed by future studies, outside the scope of the current manuscript. Consistent with the important role of calcium signaling in the regulation of erytrophagocytosis (Ma et al., 2021), we now also show that iron accumulation affects macrophage calcium levels in iRPMs (Figure 6H). Importantly, the effects of iron excess on the levels of the receptors and calcium can be rescued by DFO, similarly to overall erythrophagocytic activity (Figure 6A).

3) The proposed model is that during aging both EP and HO-1 expression decreases in RPMs but iron and ferroportin levels are elevated making for a discrepancy with elevated LIP in RPMs under these conditions.

We can only assume that a small misunderstanding in the interpretation of the presented data underlies this comment. We show that ferroportin levels in RPMs (Figure 1F) are modulated in a manner that fully reflects the iron status of these cells (both labile iron and total iron levels, Figures 1 H and I). FPN levels drop in aged RPMs and are rescued when mice are maintained on an iron-reduced diet, consistent also with the mRNA expression levels of hepcidin (Figure 1D). As pointed out by Reviewer#3, and explained in detail in our replies below (R1 Ad. 1; R3 Ad), we believe that ferroportin levels are critical for the observed phenotypes in aging.

4) Some technical elements, specifically the validation of antibodies for flow and the methods for free heme measures, require further clarification.

We now provide data as figure supplements where antibodies against HFT, LFT, FPN and HO-1 are validated for flow cytometry – they show the expected expression in response to alteration of iron levels/holo-ferritin treatment. Regarding the measurements of splenic heme levels: we fully agree with Reviewer#2 that the method is not perfect. We now clarified it better in the Methods section (p. 32) and supported these data with additional measurements, including heme levels in plasma from the portal vein (which also carries blood from the spleen, Figure 2H) and serum hemopexin levels determined by ELISA (Figure 2I).

Reviewer #2 (Recommendations for the authors):1. Since major conclusions are derived from alterations in various iron/heme metabolism protein levels, it is critical to show that these antibodies (ferritin, HO1, FPN) actually are specific and work in flow cytometry experiments. I am unware of whether these Abs were ever authenticated for flow/FACS analyses.

The Reviewer raised an important issue. As explained in detail in our reply Essential Revisions Ad. 4. we now provide the data for the validation of the antibodies against HFT, HFL, HO1 and FPN for flow cytometry measurements (Figure 1—figure supplement 3 and Figure 6—figure supplement 3). Of note, we also use the antibody against FPN in our parallel projects and we see that it performs as expected (eg. we detect high FPN levels in RPMs in iron deficiency and we see FPN downregulation upon mini-hepcidin treatment).

2. What is the underlying mechanism for lower EP? The authors propose that aging RPMs have decreased lysosomal activity, lower H ferritin, increase LIP and oxidative stress. What is the cause and effect for RPMs from aged mice to behave this way?

We believe that the primary phenotype of the aging RPMs is a progressive iron accumulation as hallmarked by an increase of both total and labile iron (Figure 1I and H). Our data in iRPMs in vitro show a causative effect of iron loading on both lysosomal and erythrophagocytic activity (Figures 6A and C). As described in the reply Essential Revisions Ad. 2b, we now provide more extensive new data demonstrating that iron loading decreases the levels of apoptotic cell receptors (MERTK, AXL, and TIM4) and of calcium levels (implicated in erythrophagocytic activity as reported by Ma et al. (Ma et al., 2021) (new Figure 6E-H)).

3. Figure 2: FPN is high in aged RBCs while Hamp is high in the liver. What is the explanation for this phenomenon? Does BMP6 mRNA increase in the liver too? Or is high Hamp level the result of low-grade inflammation (in aged mice) and consequently the activation of the Stat3 pathway?

We measured the protein levels of FPN in splenic RBCs to exclude the possibility that their high ROS levels are underlain by low FPN and hence excessive intracellular iron, as shown in erythrocyte-specific FPN KO mice (Zhang et al., 2018)(Figure 2E). We do not have a clear answer as to why these levels are even higher than in young cells, especially that RBCs lack the machinery for transcriptional control. Bmp6 mRNA levels are increased in the livers of aged mice which is now included in our independent manuscript that focuses on iron sensing mechanisms (to be submitted soon). We have not observed any signs of low-grade inflammation in aged versus young mice at the age we investigated for this manuscript. To illustrate this, we now included serum IL-6 levels in young, aged, and aged IR mice (new Figure 1E). We also observed rather a downregulation of the expression levels of proinflammatory cytokines in spleens in aged mice, and these data may be followed by our further studies.

4. Does the iron accumulation in aged RPMs due to iron accumulation in aged RBCs?

As mentioned above (R1 Ad. 5) we now provide additional data regarding RBC fitness. Consistent with the time life-span experiment (Fig, 2A), we show that oxidative stress in RBCs is only increased in splenic, but not circulating RBCs (new Figure 2C, replacing the old Figure 2B and C). In addition, we show no signs of age-triggered iron loading in RBCs, either in the spleen (new Figure 2F) or in the circulation (new Figure 2B). Hence, we do not envision a possibility that RPMs become iron-loaded during aging as a result of erythrophagocytosis of iron-loaded RBCs.

5. Page 8: Why does Fe restriction decrease ROS production in RBCs? (Figure 2CandD)

As described above (Ad. 6 to Reviewer’s#1 recommendation for the authors) we use ROS in RBCs as a readout for their senescence status. We propose that the increased proportion of high-ROS RBCs in the aged spleen is a consequence of lower RPMs number as well as diminished erythrophagocytic activity.

6. Page 8: How was heme determined and quantified? How do you know that heme is extracellular? How do you know it is "free"? (Figure 2F)

We fully agree with the Reviewer that our method has some limitations. It for sure may capture heme that is bound by proteins, such as hemoglobin or heme/hemoglobin scavengers (hemopexin or haptoglobin). We also cannot fully exclude that the measured levels come explicitly from extracellular space. Therefore, to better support our findings, we now measured heme levels in plasma derived from portal blood (that is derived also from the spleen). Consistently, we found higher levels of heme in the portal circulation from aged mice compared to young controls (new Figure 2H). In addition, we measured hemopexin levels in the serum (new Figure 2I) and we found that it follows to a large extent the levels of extracellular heme from the spleen (Figure 2G).

7. Page 8, line 42,: … our data suggest that reduced EP rate during aging may promote the splenic retention.." I would think that this means that there is higher ROS in red cells from aged mice which could pre-dispose them for preemptive clearance by RPMs, which are unhealthy to begin with in aged mice.

As mentioned above in our replies (R1 Ad. 5 and R2 Ad. 4) we did not observe any hints of decreased fitness of circulating RBCs (as indicated mainly by their normal life-span and unaltered ROS levels, Figures 2A and C). Hence, we do not expect that during aging circulating RBCs are pre-disposed to preemptive clearance.

8. Page 9, line 25: "… these aggregates contain large amounts of total iron (Figure 2J) and, interestingly, heme (Figure 2K)". The numbers (fold-change) don't match with total Fe and heme in Figure 1A, even if these were aggregates while the others are total.

Thank you for this comment. In Figure 1A we present non-heme iron content of the whole spleen. In our Figure 2M and N we show solely total iron content (likely including heme) and heme iron content of the isolated aggregates. These three pieces of data originate from different sample types and are obtained with different methods (ie, they measure different forms of iron). We assume that this likely influences the difference in fold changes. Another important factor is that splenic non-heme iron content will be also dependent on various other cell types in the spleen, whose iron status during aging is ill-understood and would require further investigations.

9. Page 12, lines 35/36: "… we propose that their formation limits the bioavailability of iron for further systemic utilization".How does this reconcile with the fact that there is 200ppm dietary Fe to overcome Fe bioavailability? Given that the authors have only looked at the spleen (and in some figures, also the liver), how does the bone marrow, the major erythropoietic organ, deal with this "limited Fe availability"?If the authors hypothesize that heme-iron deposits are insoluble thus limiting iron bioavailability for systemic utilization, then why is the anemia not corrected in aged IR mice if there is less heme-iron aggregates, and thus, presumably, more available iron?

Thanks for this comment. This is right that the standard diet contains a more-than-sufficient amount of iron. However, what we assume is that during aging a part of RBC-derived iron becomes unavailable in the spleen in the form of protein aggregates, likely due to RPMs’ damage. Concomitantly, hepcidin expression increases in the liver, and FPN drops in RPMs. We have not determined FPN in the aged duodenum, but it may also be decreased. As a consequence, if iron absorption may be mildly restricted and iron recycling is impaired, less iron is available for replenishment of the transferrin pool. Our data regarding transferrin saturation support such a model. Still, this mild iron deficiency is not enough to significantly alter all RBC parameters (see more extended data as Figure 1—figure supplement: we see a mild significant effect on Hgb, and a tendency (p=0.054) for a drop in hematocrit). The bone marrow erythropoietic activity is unaltered (Figure 1—figure supplement F). Nevertheless, the fact that EPO is mildly increased in aged mice and extramedullary erythropoiesis rate is enhanced suggests that the body senses lower iron availability/mild hypoxia in aged mice. This indeed is not corrected by the iron-reduced diet. The reason for this is not clear to us. However, as Reviewer#1 pointed out (point 3), anemia in aging is likely multifactorial, and dissection of the contributing factors (apart from iron availability) would require further studies (as we mention on p.6).

10. Page 18, line 2: "… causative link between iron loading…".But the "iron-loaded" cells in vivo have low Ftn levels. Wouldn't this mean that the reason for their perturbation is something else?

Our data show that aged RPMs accumulate large amounts of total iron and exhibit increased labile iron levels (Figure 1H and I). Our in vitro data show a causative role of iron loading in downregulating erythrophagocytosis and we now present additional insights into the candidate mechanisms (new Figure 6). We propose that likely the staining of cytoplasmic ferritin H in flow cytometry is not recognizing those ferritin particles that start to form the aggregates (in fact, extracellular aggregates were negative for flow cytometric detection of L-ferritin, even though according to MS analysis they contain large amounts of both L and H ferritin). Thus, we think that fully biologically active ferritin may be depleted in aged RPMs, hence promoting a build-up of high labile iron levels. We now included this additional reasoning in the manuscript (p. 11).

11. The justification for use of only female mice is inadequate. We find major differences between male and female mice in iron/heme metabolism and macrophage recycling.

Thank you for this comment. Indeed, several aspects of iron homeostasis differ between males and females. The fact that C57Bl/6J females show higher iron levels in the spleen actually inspired us to study primary females in our project. Investigation of both sexes in parallel would exceed our project budget. However, we now present some aspects of RPM aging in Balb/c females (Figure 2—figure supplement 4) and we comment on C57BL/6J males (p. 9). We made it clear in the revised version of the manuscript that our observations refer to female mice (eg, in the Abstract) and that the degree of splenic aggregates formation likely depends on overall body iron burden and the functioning of the hepcidin-FPN axis (please, see also the last paragraph of the Discussion, p. 18). References

Altamura, S., Kessler, R., Grone, H. J., Gretz, N., Hentze, M. W., Galy, B., and Muckenthaler, M. U. (2014). Resistance of ferroportin to hepcidin binding causes exocrine pancreatic failure and fatal iron overload. Cell Metab, 20(2), 359-367. doi:10.1016/j.cmet.2014.07.007

Aw, D., Hilliard, L., Nishikawa, Y., Cadman, E. T., Lawrence, R. A., and Palmer, D. B. (2016). Disorganization of the splenic microanatomy in ageing mice. Immunology, 148(1), 92-101. doi:10.1111/imm.12590

Brancolini, C., and Iuliano, L. (2020). Proteotoxic Stress and Cell Death in Cancer Cells. Cancers (Basel), 12(9). doi:10.3390/cancers12092385

Clarke, J., Kayatekin, C., Viel, C., Shihabuddin, L., and Sardi, S. P. (2021). Murine Models of Lysosomal Storage Diseases Exhibit Differences in Brain Protein Aggregation and Neuroinflammation. Biomedicines, 9(5). doi:10.3390/biomedicines9050446

Duez, J., Holleran, J. P., Ndour, P. A., Pionneau, C., Diakite, S., Roussel, C.,... Buffet, P. A. (2015). Mechanical clearance of red blood cells by the human spleen: Potential therapeutic applications of a biomimetic RBC filtration method. Transfus Clin Biol, 22(3), 151-157. doi:10.1016/j.tracli.2015.05.004

Dupuis, L., Chauvet, M., Bourdelier, E., Dussiot, M., Belmatoug, N., Le Van Kim, C.,... Franco, M. (2022). Phagocytosis of Erythrocytes from Gaucher Patients Induces Phenotypic Modifications in Macrophages, Driving Them toward Gaucher Cells. Int J Mol Sci, 23(14). doi:10.3390/ijms23147640

Friedmann Angeli, J. P., Schneider, M., Proneth, B., Tyurina, Y. Y., Tyurin, V. A., Hammond, V. J.,... Conrad, M. (2014). Inactivation of the ferroptosis regulator Gpx4 triggers acute renal failure in mice. Nat Cell Biol, 16(12), 1180-1191. doi:10.1038/ncb3064

Jenkitkasemwong, S., Wang, C. Y., Coffey, R., Zhang, W., Chan, A., Biel, T.,... Knutson, M. D. (2015). SLC39A14 Is Required for the Development of Hepatocellular Iron Overload in Murine Models of Hereditary Hemochromatosis. Cell Metab, 22(1), 138-150. doi:10.1016/j.cmet.2015.05.002

Kohyama, M., Ise, W., Edelson, B. T., Wilker, P. R., Hildner, K., Mejia, C.,... Murphy, K. M. (2009). Role for Spi-C in the development of red pulp macrophages and splenic iron homeostasis. Nature, 457(7227), 318-321. doi:10.1038/nature07472

Lefebvre, T., Reihani, N., Daher, R., de Villemeur, T. B., Belmatoug, N., Rose, C.,... Karim, Z. (2018). Involvement of hepcidin in iron metabolism dysregulation in Gaucher disease. Haematologica, 103(4), 587-596. doi:10.3324/haematol.2017.177816

Liu, Z., Gu, Y., Chakarov, S., Bleriot, C., Kwok, I., Chen, X.,... Ginhoux, F. (2019). Fate Mapping via Ms4a3-Expression History Traces Monocyte-Derived Cells. Cell, 178(6), 1509-1525 e1519. doi:10.1016/j.cell.2019.08.009

Ma, S., Dubin, A. E., Zhang, Y., Mousavi, S. A. R., Wang, Y., Coombs, A. M.,... Patapoutian, A. (2021). A role of PIEZO1 in iron metabolism in mice and humans. Cell, 184(4), 969-982 e913. doi:10.1016/j.cell.2021.01.024

Mascaro, M., Alonso, E. N., Alonso, E. G., Lacunza, E., Curino, A. C., and Facchinetti, M. M. (2021). Nuclear Localization of Heme Oxygenase-1 in Pathophysiological Conditions: Does It Explain the Dual Role in Cancer? Antioxidants (Basel), 10(1). doi:10.3390/antiox10010087

Okreglicka, K., Iten, I., Pohlmeier, L., Onder, L., Feng, Q., Kurrer, M.,... Kopf, M. (2021). PPARgamma is essential for the development of bone marrow erythroblastic island macrophages and splenic red pulp macrophages. J Exp Med, 218(5). doi:10.1084/jem.20191314

Ryan S.K, Zelic M., Han Y., Teeple E., Chen L., Sadeghi M.,... T.R., R. H. (2021). Microglia ferroptosis is prevalent in neurodegenerative disease and regulated by SEC24B. BioRxiv, doi: https://doi.org/10.1101/2021.11.02.466996

Slusarczyk, P., and Mleczko-Sanecka, K. (2021). The Multiple Facets of Iron Recycling. Genes (Basel), 12(9). doi:10.3390/genes12091364

Stefanova, D., Raychev, A., Deville, J., Humphries, R., Campeau, S., Ruchala, P.,... Bulut, Y. (2018). Hepcidin Protects against Lethal *Escherichia coli* Sepsis in Mice Inoculated with Isolates from Septic Patients. Infect Immun, 86(7). doi:10.1128/IAI.00253-18

Sui X., Miguel A., Prado M.A., Paulo J.A., Gygi S.P., Finley D., and R.I, M. (2022). Global proteome metastability response in isogenic animals to missense mutations and polyglutamine expansions in aging. BioRxiv, doi: https://doi.org/10.1101/2022.09.28.509812

Theurl, I., Hilgendorf, I., Nairz, M., Tymoszuk, P., Haschka, D., Asshoff, M.,... Swirski, F. K. (2016). On-demand erythrocyte disposal and iron recycling requires transient macrophages in the liver. Nat Med, 22(8), 945-951. doi:10.1038/nm.4146

Turner, V. M., and Mabbott, N. A. (2017). Influence of ageing on the microarchitecture of the spleen and lymph nodes. Biogerontology, 18(5), 723-738. doi:10.1007/s10522-017-9707-7

Youssef, L. A., Rebbaa, A., Pampou, S., Weisberg, S. P., Stockwell, B. R., Hod, E. A., and Spitalnik, S. L. (2018). Increased erythrophagocytosis induces ferroptosis in red pulp macrophages in a mouse model of transfusion. Blood, 131(23), 2581-2593. doi:10.1182/blood-2017-12-822619

Zhang, D. L., Wu, J., Shah, B. N., Greutelaers, K. C., Ghosh, M. C., Ollivierre, H.,... Rouault, T. A. (2018). Erythrocytic ferroportin reduces intracellular iron accumulation, hemolysis, and malaria risk. Science, 359(6383), 1520-1523. doi:10.1126/science.aal2022

[Editors’ note: further revisions were suggested prior to acceptance, as described below.]

1) While the authors explained clearly in the response to reviewers regarding point 1 of the essential revision, the explanation does not appear evident or as clear in the revised manuscript. Can the authors note where this is in the abstract, introduction, or discussion? There should be a sentence somewhere in the abstract or discussion or both that says "inflammation with aging leads to increasing hepcidin which suppresses ferroportin, leading to iron accumulation, aggregate formation, and RPM depletion/dysfunction."

We have now included a new sentence in the Abstract, pointing to the drop in ferroportin as the most likely cause for intracellular iron loading of RPMs (page 2, line 39-40). The fragment of the Introduction that contains this information is on page 4 (line 119-120) and in the Discussion on page 16 (line 544-546). As we show in our data in Figure 1E, the inflammatory cues are rather unlikely to cause an increase in hepcidin and a drop in ferroportin, and therefore we would prefer not to include ‘inflammation’ as the factor. Of note, this is also consistent with rather an anti-inflammatory not a pro-inflammatory transcriptional profile of aged RPMs in the RNAseq data.

Furthermore, aggregates correlate with RPM loss in which the only conclusion is that monocyte recruitment does not compensate sufficiently for RPM loss with aging. Please modify in the Discussion on Page 16.

We tried to modify the indicated fragment of the Discussion for better clarity (now on page 15, lines 522-529). Our point was that similar to the aging model, also the other conditions/challenges mentioned in the text are hallmarked by RPM depletion (for some of them it is not clear from the published data if the reduction of RPM numbers would be transient or sustained). Nevertheless, what we want to propose is that independent of the replenishment of the niche, damaged RPMs may give rise to the formation of iron-rich protein aggregates in the models mentioned, a possibility that was not addressed by previous studies.

We hope that overall our data are convincing to show that RPMs are the major source of the aggregates.

The authors are hedging through the manuscript about what is the primary cause and what effect, and it is unclear what causes the initial inflammatory trigger (maybe just accumulated inefficiency of EP?).

Based on the first round of revision, we understood that assigning causality to our observations is important and we followed this suggestion with additional experiments and argumentation in the text. We attempted to understand the primary cause for iron loading in RPMs, and we show that it might be attributable to ferroportin downregulation. We propose that it is linked to hepcidin upregulation, but we do not assign this regulation to ‘inflammaging’ (Figure 1E). As mentioned in our replies to the previous round of revision, we do see that Bmp6 expression is also increased in aging, but we characterize this response in an independent manuscript.

Removing some of the data and discussion that is afield and focusing on the main thrust of this manuscript would help enormously with readability. 6 figures with 8-16 panels each detract from understanding the main points. Please edit (suggestions made here and below).

We now removed some data from the manuscript or included them in the supplement, following the suggestions below. We hope that these efforts improved the readability of our manuscript.

2) There appears to be a discrepancy between ferritin decrease with aging and the expectation that it is increased in inflammation, aging, and Figure 7. The authors note that they think ferritin is aggregated but labile iron in RBCs is increased in the spleen from IR-aged mice (Figure 2F). Please edit Figure 7 accordingly, explain the findings in Figure 2F, and add to the Discussion that how ferritin aggregation is affected by excess iron during aging remains to be explored.

We now modified Figure 7 to illustrate a drop in ferritin, which is paralleled by the formation of iron-rich aggregates that contain large amounts of ferritin (based on our MS data), as we write in the manuscript (page 11, line 353-356). We think that high amounts of labile iron in aged RPMs appear mostly due to a drop in ferroportin, as only the ferroportin expression on RPM membranes, but not ferritin levels or protein aggregation are rescued by the IR diet.

Still, deficient levels of fully-functional ferritin likely contribute to the labile iron build-up. We now added the sentence suggested by the Reviewer to the Discussion (page 16, line 545-546)

Regarding the previous Figure 2F, we measured labile iron in splenic RBCs in response to the comments that we received from the Reviewers. The aim of this was to show that RPMs do not become iron loaded in aging because of EP of iron-loaded RBCs. We do not see a direct connection between the ferritin status of RPMs and labile iron in RBC. Following the suggestions below, we now moved these data to Supplements and commented that the increased ROS status in splenic RBCs reflects their senescence and is not underlain by FPN drop or iron accumulation.

3) IR diet does not rescue anemia or alter RNA seq data and the use of agents to remove macrophages altogether does not worsen anemia in mouse models of the disease. Taken together, this raises doubt regarding the central role of RPMs in anemia of aging.

In our manuscript we do not emphasize the anemia of aging, as at least in the studied mice at the given age the anemic phenotype is very mild. We focus on the phenotypic changes of RPMs in response to increased iron levels in the context of aging, which are novel and we believe may be of broader relevance. We think that a mild drop in transferrin saturation, which we partially rescue by the IR diet, and which at least to some extent may be caused by less efficient iron recycling by RPMs, is more important and in principle may affect as well other cell types than only erythroid cells.

While the data is strong overall, there is a possibility that the findings are related more to the standard mouse chow which is a super-physiological iron diet. This must be addressed as a weakness of the study since what the authors actually demonstrate is a comparison between the long-term effects of an iron overload diet and a normal iron diet on RPMs. The manuscript can be modified toward this conceptually or include this idea in the Discussion section since it may not be relevant for normal aging.

The Reviewer raised a valid issue. Modifying the whole manuscript conceptually would be very difficult at this stage. We now included the suggested idea in the Discussion (page 17, line 588-590).

References

Bennett, L. F., Liao, C., Quickel, M. D., Yeoh, B. S., Vijay-Kumar, M., Hankey-Giblin, P.,... Paulson, R. F. (2019). Inflammation induces stress erythropoiesis through heme-dependent activation of SPI-C. Sci Signal, 12(598). doi:10.1126/scisignal.aap7336

Bian, Z., Shi, L., Guo, Y. L., Lv, Z., Tang, C., Niu, S.,... Liu, Y. (2016). Cd47-Sirpalpha interaction and IL-10 constrain inflammation-induced macrophage phagocytosis of healthy self-cells. Proc Natl Acad Sci U S A, 113(37), E5434-5443. doi:10.1073/pnas.1521069113

Boretti, F. S., Baek, J. H., Palmer, A. F., Schaer, D. J., and Buehler, P. W. (2014). Modeling hemoglobin and hemoglobin:haptoglobin complex clearance in a non-rodent species-pharmacokinetic and therapeutic implications. Front Physiol, 5, 385. doi:10.3389/fphys.2014.00385

Etzerodt, A., Kjolby, M., Nielsen, M. J., Maniecki, M., Svendsen, P., and Moestrup, S. K. (2013). Plasma clearance of hemoglobin and haptoglobin in mice and effect of CD163 gene targeting disruption. Antioxid Redox Signal, 18(17), 2254-2263. doi:10.1089/ars.2012.4605

Haldar, M., Kohyama, M., So, A. Y., Kc, W., Wu, X., Briseno, C. G.,... Murphy, K. M. (2014). Heme-mediated SPI-C induction promotes monocyte differentiation into iron-recycling macrophages. Cell, 156(6), 1223-1234. doi:10.1016/j.cell.2014.01.069

Lu, Y., Basatemur, G., Scott, I. C., Chiarugi, D., Clement, M., Harrison, J.,... Mallat, Z. (2020). Interleukin-33 Signaling Controls the Development of Iron-Recycling Macrophages. Immunity, 52(5), 782-793 e785. doi:10.1016/j.immuni.2020.03.006

Ma, S., Dubin, A. E., Zhang, Y., Mousavi, S. A. R., Wang, Y., Coombs, A. M.,... Patapoutian, A. (2021). A role of PIEZO1 in iron metabolism in mice and humans. Cell, 184(4), 969-982 e913. doi:10.1016/j.cell.2021.01.024

Pek, R. H., Yuan, X., Rietzschel, N., Zhang, J., Jackson, L., Nishibori, E.,... Hamza, I. (2019). Hemozoin produced by mammals confers heme tolerance. eLife, 8. doi:10.7554/eLife.49503

Schaer, D. J., Schaer, C. A., Buehler, P. W., Boykins, R. A., Schoedon, G., Alayash, A. I., and Schaffner, A. (2006). CD163 is the macrophage scavenger receptor for native and chemically modified hemoglobins in the absence of haptoglobin. Blood, 107(1), 373-380. doi:10.1182/blood-2005-03-1014

Youssef, L. A., Rebbaa, A., Pampou, S., Weisberg, S. P., Stockwell, B. R., Hod, E. A., and Spitalnik, S. L. (2018). Increased erythrophagocytosis induces ferroptosis in red pulp macrophages in a mouse model of transfusion. Blood, 131(23), 2581-2593. doi:10.1182/blood-2017-12-822619